# Constrained Decoding for Cross-lingual Label Projection

**Duong Minh Le, Yang Chen, Alan Ritter & Wei Xu**
Georgia Institute of Technology
{dminh6, yangc}@gatech.edu,{alan.ritter, wei.xu}@cc.gatech.edu

## Abstract

Zero-shot cross-lingual transfer utilizing multilingual LLMs has become a popular learning paradigm for low-resource languages with no labeled training data. However, for NLP tasks that involve fine-grained predictions on words and phrases, the performance of zero-shot cross-lingual transfer learning lags far behind supervised fine-tuning methods. Therefore, it is common to exploit translation and label projection to further improve the performance by (1) translating training data that is available in a high-resource language (e.g., English) together with the gold labels into low-resource languages, and/or (2) translating test data in low-resource languages to a high-source language to run inference on, then projecting the predicted span-level labels back onto the original test data. However, state-of-the-art marker-based label projection methods suffer from translation quality degradation due to the extra label markers injected in the input to the translation model. In this work, we explore a new direction that leverages constrained decoding for label projection to overcome the aforementioned issues. Our new method not only can preserve the quality of translated texts but also has the versatility of being applicable to both translating training and translating test data strategies. This versatility is crucial as our experiments reveal that translating test data can lead to a considerable boost in performance compared to translating only training data. We evaluate on two cross-lingual transfer tasks, namely Named Entity Recognition and Event Argument Extraction, spanning 20 languages. The results demonstrate that our approach outperforms the state-of-the-art marker-based method by a large margin and also shows better performance than other label projection methods that rely on external word alignment[1].

## 1 Introduction

Large language models (LLMs) have demonstrated the potential to perform a variety of NLP tasks in zero or few-shot learning settings. This is attractive because labeling data is expensive — annotating fine-tuning data across many languages for each task is not feasible. However, for traditional NLP tasks that involve word/phrase-level predictions, such as named entity recognition or event extraction, the performance of zero and few-shot learning lags far behind supervised fine-tuning methods that make use of large amounts of labeled data (Lai et al., 2023). Prior work has therefore trained multilingual models that support cross-lingual transfer from a high-resource language (e.g., English), where fine-tuning data is available to many low-resource languages where data may not be available (e.g., Bambara, which is spoken primarily in Africa). Encoder-based LLMs such as XLM-RoBERTa (Conneau et al., 2020) or mDeBERTa (He et al., 2021) work surprisingly well for cross-lingual transfer, yet the performance of models that are fine-tuned on target-language data is still significantly better (Xue et al., 2021). Motivated by this observation, we present a new approach to automatically translate NLP training datasets into many languages that uses constrained decoding to more accurately translate and project annotated label spans from high to low-resource languages.

Our approach builds on top of EasyProject (Chen et al., 2023a), a simple, yet effective state-of-the-art method for label projection, that inserts special markers (see Figure 1a) into the source sentences

---

[1]Our code is available at: https://github.com/duonglm38/Codec

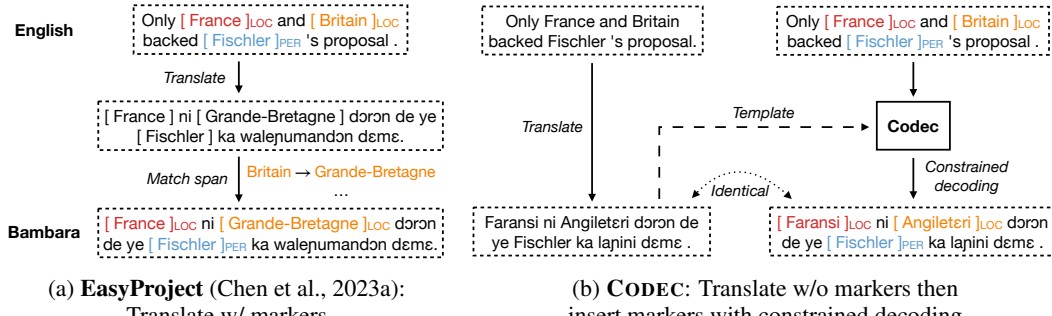

(a) **EasyProject** (Chen et al., 2023a):
Translate w/ markers

(b) **CODEC**: Translate w/o markers then
insert markers with constrained decoding

Figure 1: The goal of label projection is to automatically construct annotated data in low-resource language (e.g., Bambara) by translating from annotated data in high-resource language (e.g., English) while preserving the span-level labels. **EasyProject** (Left): The source sentence is first injected with marker pairs around entities and then translated to the target language; a span-matching step is performed to map each span to the corresponding label (i.e., types of entities). This method has issues related to the translation quality (e.g., the word "France" is not properly translated), due to the existence of markers in the input to the translation model. **CODEC** (Right): The source sentence is first translated to the target language, given the source sentence without marker injected, CODEC then performs constrained decoding to insert markers to the translated sentence.

to mark annotated spans, then runs the modified sentences through a machine translation (MT) system, such as NLLB (Costa-jussà et al., 2022) or Google Translate. A key limitation of EasyProject, as noted by Chen et al. (2023a), is that inserting special markers into the source sentence then translating it degrades the translation quality; nevertheless, EasyProject was shown to be more effective than prior work for label projection that largely relied on word alignment (Yarmohammadi et al., 2021). To address the problem of translation quality degradation, in this paper, we present a new approach, **Co**nstraint **De**coding for **C**ross-lingual Label Projection (CODEC), for translating training datasets using a customized constrained decoding algorithm. The training data in the high-resource language is first translated *without markers* followed by a second constrained decoding pass to inject the markers (see Figure 1b). Since the source sentence does not include markers during the translation phase, the final translated text quality from CODEC is preserved. The second decoding pass of CODEC relies on a translation model that is conditioned on the modified input sentence *with markers* (thus is noisier) in order to find the appropriate positions for inserting markers. Using a specially designed constrained decoding algorithm, however, we can retain the high-quality translation while having the right number of labels projected by enforcing both as constraints during decoding.

In essence, CODEC only explores the search space which contains valid hypotheses, i.e., translated outputs that conform to (i) the high-quality translation from the first decoding pass without markers' interference and (ii) having the correct number of markers inserted. A brute-force enumeration of all possible such hypotheses is intractable, as the number of sequences that would need to be scored using the translation model is $O(n^{2m})$, where $n$ is the sequence length and $m$ is the number of labeled spans to be projected, as we show in §3. We therefore design a constrained decoding algorithm based on the branch-and-bound method (Stahlberg & Byrne, 2019), in which a depth-first search is conducted to identify a lower-bound on the best complete hypothesis, and branches that do not have any solutions with a better score than the current lower bound are pruned from the search space. However, even when pruning branches using this bound, decoding time is still prohibitively long. To speed up decoding, we introduce a new *heuristic lower bound*, which removes branches more aggressively. We also introduce a technique to prune unlikely positions for the opening markers in advance. Putting everything together, compared to exact branch-and-bound search, our proposed method significantly reduces decoding time with only a slight drop in performance in a few languages. For example, for the Bambara language, CODEC is about 60 times faster than exact search, while only losing 0.6 absolute F1, a 1.1% drop in performance.

We conduct extensive experiments to evaluate CODEC on two popular cross-lingual tasks (i.e., Named Entity Recognition and Event Argument Extraction), covering 20 language pairs. In our experiment, CODEC and other label projection baselines are used to project the label from English datasets to their translated version to augment the data in the target language, which is referred to as *translate-train* (Hu et al., 2020). The results demonstrate that, on average, the model fine-tuned

on CODEC-augmented data outperforms models fine-tuned on the data produced by other label projection baselines by a large margin. This reinforces our hypothesis that preserving translation quality is essential and constrained decoding can improve the accuracy of label projection. Moreover, since CODEC separates two phases of translation and marker insertion, it can also be used to improve cross-lingual transfer by using machine translation at inference time, sometimes referred to as *translate-test* (Artetxe et al., 2023). This approach translates test data from the low-resource language to the high-resource language, uses a fully-supervised NLP model to automatically annotate the translation on the high-resource side, then projects the annotations back to the original language. Experiments show that, compared to translate-train alone, using CODEC in the translate-test setting further boosts cross-lingual transfer performance in the Named Entity Recognition task.

## 2 LABEL PROJECTION AS CONSTRAINED TRANSLATION

In this section, we describe how we formulate Label Projection as a constrained generation problem before proposing to tackle it by designing a constrained decoding algorithm (detailed in §3). Given a source sentence with label spans and its translation in the target language, our goal is to map the label spans from the source to the target sentence. Formally, let $x$ be the source sentence with $m$ labeled spans (e.g., $m$ named entities) and $x^{mark}$ be the same text as $x$ but with $m$ pairs of special markers (e.g. square brackets) surrounding every label span. In previous work, an MT model will find the translation $y^{mark}$ of the sentence with markers inserted, $x^{mark}$:

$$y^{mark} = \arg\max_y \log P_\tau(y|x^{mark}) \tag{1}$$

where $\tau$ is a machine translation (MT) system. However, as previous work has found that inserting markers degrades translation quality (Chen et al., 2023a), we introduce a variant of the approach. Our approach uses the translation of the source sentence without markers, which will act as a *template* to be injected with markers later to create the annotated sentence in the target language:

$$y^{tmpl} = \arg\max_y \log P_\tau(y|x) \tag{2}$$

In general, without the interference of special markers in the input, the translation $y^{tmpl}$ is expected to have higher quality than $y^{mark}$, as shown in (Chen et al., 2023a). Let $\mathcal{Y}$ be the set of all possible valid hypotheses with $m$ marker pairs injected into $y^{tmpl}$. In our work, we do not consider the case of overlapped or nested label spans, thus the size of $\mathcal{Y}$ is $\binom{n+2m}{2m}$ or $O(n^{2m})$, where $n$ is the length (# of tokens) of $y^{tmpl}$.

Our goal is to find the best hypothesis $y^*$ from $\mathcal{Y}$, in which all the markers are inserted at correct positions into $y^{tmpl}$ with each span found in $y^*$ mapped to its corresponding label in $x^{mark}$. We can cast this task as a constrained translation problem, which enforces two constraints: (i) $y^*$ contains exactly $m$ valid pairs of markers as $x^{mark}$ (i.e., no mismatched brackets) and (ii) the plain text (with all markers removed) of $y^*$ is the exactly the same as $y^{tmpl}$. Consequently, we can solve this problem by designing a specialized constrained decoding algorithm, which explores every hypothesis in the search space $\mathcal{Y}$ and finds the one with the highest generative probability:

$$y^* = \arg\max_{y \in \mathcal{Y}} \log P_\tau(y|x^{mark}; y^{tmpl}) \tag{3}$$

$$\log P_\tau(y|x^{mark}; y^{tmpl}) = \sum_{i=1}^n \log P_\tau(y_i|y_{<i}, x^{mark}; y^{tmpl}) \tag{4}$$

Starting with a translation prefix $\epsilon$ (e.g., a language code `<bam>`), the decoding algorithm will iteratively expand the hypothesis; when the end-of-sentence token (i.e., ``) is generated, a candidate projection of the labeled spans is found.

## 3 CONSTRAINED DECODING

In this section, we will propose a constrained decoding algorithm, CODEC, particularly designed for cross-lingual transfer learning. It uses approximation to reduce the computational complexity of the original problem and a ranking method to identify hypotheses with accurate span projections. CODEC has three main steps: (1) prune all unlikely opening-marker positions, (2) search for k hypotheses with the highest probability, and (3) re-rank to find the best hypothesis.

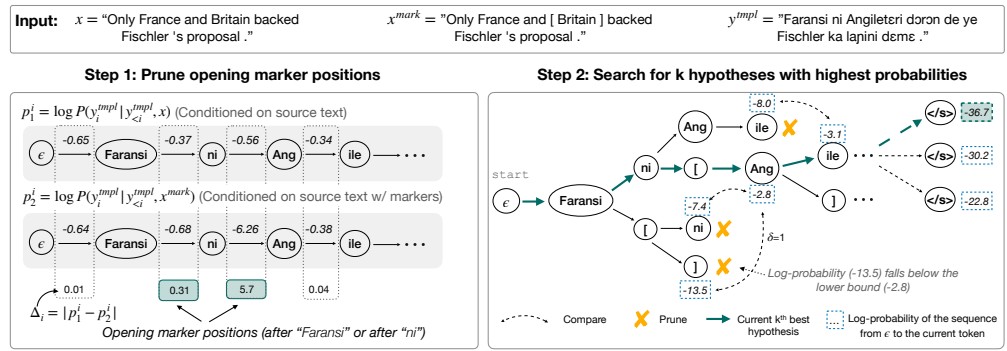

Figure 2: The first two steps of CODEC. **Step 1** (Left): CODEC prunes search branches based on unlikely opening marker positions in the target language by comparing probabilities conditioned on the source language with and without span markers inserted. **Step 2** (Right): A branch-and-bound search algorithm is used to find $k$ hypotheses with the highest probabilities ($k = 3$). From each node of the search tree (e.g., "Faransi"), CODEC expands to the next token from the translation template $y^{tmpl}$ (e.g., "ni") or a marker (i.e., "[" or "]"). A search branch is pruned if its score falls below a heuristic lower bound. Two branches of different lengths might have different values of lower bound (e.g., the lower bound for the top and bottom branches are -3.1 and -2.8, respectively).

## 3.1 PROBLEM APPROXIMATION

As discussed in §2, if we naively project $m$ label spans into the template translation $y^{tmpl}$, the research space $\mathcal{Y}$ has a size of $O(n^{2m})$ where $n$ is the number of tokens in $y^{tmpl}$. Therefore, instead of solving this $m$-projection problem, we propose to address $m$ 1-projection problems, whose search space is only $\binom{n+2}{2}$ or $O(n^2)$ each, and approximate the original solution by combining the solutions from the m problems. In other words, we will project $m$ sentences, each of which only contains one label span surrounded by one marker pair. This approximation also brings another merit, since there is only one span projected at a time, we can identify the label of the span (e.g., types of entities) on the target language without the need for the label matching step as in EasyProject (see Figure 1a).

## 3.2 PRUNING OPENING-MARKER POSITIONS

To further reduce the search space and speed up the decoding algorithm, we propose a heuristic method to detect which position in the translation template can be inserted with the opening marker (e.g., left square bracket '['). During the search process, all hypotheses that have the opening marker inserted in other positions will get pruned. The key idea of our method is to track the difference in conditional word (more accurately subword token) probabilities of generating the same translation template, when being conditioned on two different inputs: the source sentence with and without markers. The intuition is that, if we decode the translation template but conditioned on the marker sentence, at the position that needs to be inserted an opening marker, the model would give a high probability to this token, and consequently, it would assign a low probability to the token from the template. This intuition is illustrated in Figure 2 (Step 1). Formally, we define $\Delta_i$ as the difference of the two log-probabilities when generating the $i^{th}$ token from the template:

$$\Delta_i = |\log P(y_i^{tmpl}|y_{<i}^{tmpl}, x^{mark}) - \log P(y_i^{tmpl}|y_{<i}^{tmpl}, x)| \tag{5}$$

where $y_{<i}^{tmpl}$ indicates all tokens before the $i^{th}$ token and, from hereon, $P$ is used in place of $P_\tau$ for simplicity. We determine the possible positions of the opening marker by choosing all $i$ with a high value of $\Delta_i$ (i.e., larger than a threshold). All hypotheses, which have the opening marker inserted at other positions, are pruned during the search.

## 3.3 SEARCHING FOR TOP-$k$ HYPOTHESES

According to our preliminary study, the hypothesis with the highest probability in many cases is not necessarily the best output, i.e., the one with markers injected correctly. This is typically called

*(translation) model error* in machine translation research, in contrast to *search error*, where the highest scored sequence is a good translation but the decoding algorithm (often approximate search) fails to find it. Previous research on translation decoding (Stahlberg & Byrne, 2019; Eikema & Aziz, 2020) also found that model errors occur very often when using the exact search.

To address this problem, we propose to first find top-$k$ hypotheses with the highest probabilities, then use a ranking function (details in §3.4) to choose the best one (i.e., re-ranking). The main idea is to traverse the entire search space in a depth-first order and adopt the branch-and-bound method (Stahlberg & Byrne, 2019) to reduce the decoding time — see Figure 2 for an illustration. Compared to the search algorithm in the previous work (Stahlberg & Byrne, 2019), CODEC uses a different search strategy to enforce additional constraints for label projection. More specifically, in order for the generated text to follow the translation template, at each decoding iteration, the decoding scheme only considers at most two candidates for the next token: the next token from the template and the next span marker. Besides the search strategy, CODEC also introduced a new heuristic lower bound. Before presenting the lower bound of CODEC, we first discuss the one introduced by Stahlberg & Byrne (2019) and the reason why we need to define a new lower bound for this task. The purpose of the lower bound is to prune branches that do not have any solutions with a better score. The score of a branch, which has expanded a sequence to the length of $j$ (i.e., $y_{1:j}$), is the generative log-probability of this sequence (i.e., $\log P(y_{1:j}|x^{mark}) = \sum_{i=1}^{j} \log P(y_i|y_{<i}, x^{mark})$). Previous work has defined an exact lower bound as the log-probability of the current best hypothesis. Generally, let $y^k$ be the hypothesis whose log-probability is the $k^{th}$ highest at a specific time during the search ($k = 1$ when only need to find the best hypothesis). Let $L^k$ be a list containing generative log-probabilities of all prefix of $y^k$ (i.e., $L_j^k = \log P(y_{1:j}^k|x^{mark})$). The exact lower bound is defined as:

$$\gamma^{exact} = L_{|y^k|}^k \tag{6}$$

One problem with the exact lower bound is that it uses the probability of a complete hypothesis (i.e., the hypothesis ends with ``) to define the lower bound. This probability is often small, and thus cannot prune off short partial hypotheses early enough. As illustrated in Figure 2 (Step 2), the exact lower bound with $k = 3$ at this search stage is $-36.7$, much smaller than the log-probabilities of other expanded partial hypotheses. Therefore, we propose a heuristic lower bound, whose value changes according to the length of each partial hypothesis (i.e., having a larger value when compared to shorter partial hypotheses), to help expedite the pruning. The heuristic lower bound for a partial hypothesis of length $j$ is defined as:

$$\gamma = L_d^k \tag{7}$$
$$d = \min\left(\max\left(j + \delta, q\right), |y^k|\right) \tag{8}$$

where $q$ is the position of the opening marker in $y^k$ ($q = 0$ if the marker is not selected yet) and $\delta$ is a hyperparameter of the lower bound. The lower bound with a larger value of $\delta$ is closer to the exact lower bound. With the new lower bound, CODEC can prune off the expanding hypotheses as shown in Figure 2 (Step 2).

### 3.4 RE-RANKING

The goal of this step is to pick the best hypotheses, which have markers injected in the correct positions, among the top-$k$ candidates found in the previous step (§3.3). Two scores are used for this purpose: (i) hypothesis-level score: the generative log-probability of a hypothesis (Eq 4) and (ii) span-level score: the log-probability of generating the original span given the label span found in a hypothesis. In particular, for a label span $e^{src}$ in the source sentence, we have $k$ hypotheses of projecting $e^{src}$ to the target language after the search step. Let $e_i^{tgt}$ be the label span found in the i$^{th}$ hypothesis. The span-level score of the hypothesis i$^{th}$ is defined as:

$$S_i^{span} = \log P(e^{src}|e_i^{tgt}) \tag{9}$$

The same MT model is used to compute both hypothesis-level and span-level scores. The $k$ hypotheses are first ranked by the hypothesis-level score. The span-level score is then used to re-rank all hypotheses, whose label spans are equal to or are subsequences of the label span of the current top-1 hypothesis. The best hypothesis is the new top-1 after the re-ranking. More details about CODEC can be found in the Appendix §A.

## 4  EXPERIMENTS

We evaluate CODEC on two benchmark cross-lingual NLP tasks, including Named Entity Recognition and Event Argument Extraction, covering 20 languages in total.

### 4.1  DATASETS

For Named Entity Recognition (NER), we use English CoNLL03 (Tjong Kim Sang, 2002) as train/dev data and use MasakhaNER2.0 (Adelani et al., 2022) which consist of human-labeled data for 20 African languages as test data. Following Chen et al. (2023a), we consider three main types of entities (i.e., person, organization, and location) in the dataset and skip Ghomala and Naija (two languages that NLLB does not support). We conduct experiments on MasakhaNER2.0, instead of WikiANN (Pan et al., 2017), as the latter was automatically constructed dataset and contains a higher level of noises (Lignos et al., 2022). For Event Argument Extraction (EAE), we use the ACE-2005 (Doddington et al., 2004), a multilingual dataset that covers English, Chinese, and Arabic. The task is to identify the event arguments and their roles in the text given the event triggers. We use the English training sets of CoNLL03 (Tjong Kim Sang, 2002) and ACE-2005 to act as the source language to perform cross-lingual transfer for the NER and EAE tasks, respectively.

### 4.2  EXPERIMENTAL SETTINGS AND RESULTS

**Setup**  We use NLLB (No Language Left Behind) as the translation model (Costa-jussà et al., 2022) in our experiments. We fine-tune mDeBERTa-v3 (276M) to act as a NER tagger following (Chen et al., 2023a); and fine-tune mT5-large (Xue et al., 2021) following the X-Gear framework (Huang et al., 2022) for EAE. For a direct comparison with existing work (Chen et al., 2023b;a; Huang et al., 2022), we report the average F1 scores across five random seeds for NER and three random seeds for EAE. More details are provided in the Appendix §B.

**Baselines**  We compare CODEC with the following baselines: (1) **EasyProject** (Chen et al., 2023a): a state-of-the-art marker-based label projection approach; (2) **Awes-align**: an alignment-based label projection approach, in which the state-of-the-art word aligner (Dou & Neubig, 2021) is used to align the text spans between the source and target sentences; (3) **$FT_{En}$**: the multilingual models (the ones described in "Setup" above) fine-tuned directly on the English data, which have shown to be a very strong baseline for cross-lingual transfer; (4) **GPT-4**: in the cross-lingual NER task, we also prompt GPT-4-0613 (Achiam et al., 2023) to annotate each word using BIO scheme, following Chen et al. (2023b). Due to cost constraints, we only evaluate GPT-4 on 200 examples for each language of MasakhaNER2.0. EasyProject uses an NLLB model fine-tuned on a synthetic dataset as the MT model, in which the marker pairs are inserted around the label spans in both source and target sentences. We also use the fine-tuned version of NLLB-600M that they provide in CODEC for a more fair comparison.

**Results**  Table 1 shows the performance of different approaches on the test set of MasakhaNER2.0. On average, label projection methods outperform both GPT-4 and $FT_{En}$ by a large margin. Among the label projection methods, CODEC achieves the best results in both translate-train and translate-test settings, on average. Compared to EasyProject, a similar method to CODEC but using a different decoding algorithm, CODEC has achieved better or comparable performance in most of the languages, with improvements of at least 1.5 F1 in nine languages. The improvement is especially evident for the chiShona language, which is +16.5 F1. Having higher translation quality may be the reason why CODEC can outperform EasyProject in many languages. One noticeable issue with the translation of EasyProject is that the label spans are more often left untranslated (similar to the example in Figure 1a). Other types of error from EasyProject and Awes-align are demonstrated in Figure 4 in the Appendix. In addition, we observe that cross-lingual NER in African languages favors translate-test approaches more than translate-train overall. Compared to when being used in the translate-train setting, both CODEC and Awes-align have significant increases in the translate-test. In the translate-test setting, CODEC also shows better results than the latter method in 13 out of 18 languages. Another interesting observation is that $FT_{En}$ achieves the best performance in Chichewa and Kiswahili, likely because the entities in these two languages are often kept the same as their English form. Therefore, fine-tuning the NER model on English data only is sufficient for the two languages

Table 1: Cross-lingual NER results on the test set of MasakhaNER2.0 (the best F1 score of a language is in **bold**, $\Delta_{FT}$: calculated against $FT_{En}$, †: GPT-4 is evaluated on a subset of 200 examples from the test set of each language due to cost constraints). On average, CODEC outperforms other approaches by a large margin especially when used with the translate-test strategy; EasyProject is not applicable to translate-test.

| Lang. | GPT-4[†] | $FT_{En}$ | Translate-train | | | Translate-test | |
|---|---|---|---|---|---|---|---|
| | | | Awes-align | EasyProject | CODEC ($\Delta_{FT}$) | Awes-align | CODEC ($\Delta_{FT}$) |
| Bambara | 46.8 | 37.1 | 45.0 | 45.8 | 45.8 (+8.7) | 50.0 | **55.6** (+18.5) |
| Ewe | 75.5 | 75.3 | 78.3 | 78.5 | **79.1** (+3.8) | 72.5 | **79.1** (+3.8) |
| Fon | 19.4 | 49.6 | 59.3 | 61.4 | **65.5** (+15.9) | 62.8 | 61.4 (+11.8) |
| Hausa | 70.7 | 71.7 | 72.7 | 72.2 | 72.4 (+0.7) | 70.0 | **73.7** (+2.0) |
| Igbo | 51.7 | 59.3 | 63.5 | 65.6 | 70.9 (+11.6) | **77.2** | 72.8 (+13.5) |
| Kinyarwanda | 59.1 | 66.4 | 63.2 | 71.0 | 71.2 (+4.8) | 64.9 | **78.0** (+11.6) |
| Luganda | 73.7 | 75.3 | 77.7 | 76.7 | 77.2 (+1.9) | **82.4** | 82.3 (+7.0) |
| Luo | **55.2** | 35.8 | 46.5 | 50.2 | 49.6 (+13.8) | 52.6 | 52.9 (+17.1) |
| Mossi | 44.2 | 45.0 | 52.2 | 53.1 | **55.6** (+10.6) | 48.4 | 50.4 (+5.4) |
| Chichewa | 75.8 | **79.5** | 75.1 | 75.3 | 76.8 (-2.7) | 78.0 | 76.8 (-2.7) |
| chiShona | 66.8 | 35.2 | 69.5 | 55.9 | 72.4 (+37.2) | 67.0 | **78.4** (+43.2) |
| Kiswahili | 82.6 | **87.7** | 82.4 | 83.6 | 83.1 (-4.6) | 80.2 | 81.5 (-6.2) |
| Setswana | 62.0 | 64.8 | 73.8 | 74.0 | 74.7 (+9.9) | **81.4** | 80.3 (+15.5) |
| Akan/Twi | 52.9 | 50.1 | 62.7 | 65.3 | 64.6 (+14.5) | 72.6 | **73.5** (+23.4) |
| Wolof | 62.6 | 44.2 | 54.5 | 58.9 | 63.1 (+18.9) | 58.1 | **67.2** (+23.0) |
| isiXhosa | 69.5 | 24.0 | 61.7 | **71.1** | 70.4 (+46.4) | 52.7 | 69.2 (+45.2) |
| Yoruba | **58.2** | 36.0 | 38.1 | 36.8 | 41.4 (+5.4) | 49.1 | 58.0 (+22.0) |
| isiZulu | 60.2 | 43.9 | 68.9 | 73.0 | **74.8** (+30.9) | 64.1 | **76.9** (+33.0) |
| AVG | 60.4 | 54.5 | 63.6 | 64.9 | 67.1 (+12.7) | 65.8 | **70.4** (+16.0) |

in this setting. As for the EAE task, we observe a similar trend in Table 2 that CODEC outperforms the other label projection approaches on average. In each language, CODEC also has comparable or better performance than each baseline. We only report the performance in the translate-train setting for the EAE task, because the typical experiment setup requires the gold event trigger as input, but for translate-test, this information in English can only achieved by another pass of label projection (from a target language to English) during inference time. More discussions about the two cross-lingual tasks are in Appendix C, D.

## 4.3 ABLATION STUDY

In this section, we study the efficiency and accuracy of the two heuristic steps in CODEC.

**Setup** We modify each module of CODEC and evaluate the performance of each setting in translate-dev on a sample of MasakhaNER2.0 dev set for five languages (i.e., Bambara, Fon, Mossi, Yoruba, and isiZulu). We evaluate the performance of **exact (+re_rank)**: exact search algorithm, similar to CODEC except using the exact lower bound (discussed in §3) and does not prune the opening marker positions; **exact**: similar to

Table 2: F1 scores of different methods on ACE-2005 dataset in the translate-train setting.

| Lang. | $FT_{en}$ | Awes-align | EasyProject | CODEC |
|---|---|---|---|---|
| Arabic | 44.8 | 48.3 | 45.4 | **48.4** |
| Chinese | 54.0 | 57.3 | **59.7** | 59.1 |
| AVG | 49.4 | 52.8 | 52.6 | **53.8** |

exact (+re_rank) but does not conduct the re-ranking and only return hypothesis with the highest probability; **CODEC ($\delta$=1)**, **CODEC ($\delta$=3)**: CODEC with different value of $\delta$, the hyperparameter of the heuristic lower bound in Eq. (8); **CODEC ($\delta$=1+[)**, **CODEC ($\delta$=3+[)**: CODEC with different values of $\delta$ and does pruning the opening-marker positions. Since the exact search takes an extremely long time to complete, we sample only 100 examples with up to 5 label spans for each language.

**Results** The performance and decoding time of different search settings are shown in Figure 3a and Figure 3b, respectively. Firstly, compared to *exact (+re-rank)*, only returning the hypothesis with the highest probability (*exact*) resulted in a significant drop in performance in all languages. This observation shows the necessity of the re-ranking step to choose the best hypothesis. In terms of the heuristic lower bound, CODEC ($\delta$=3) has roughly the same F1 as *exact (+re-rank)*, showing that the heuristic lower bound with $\delta = 3$ is a good approximation of the exact lower bound while having a much better decoding speed. Finally, pruning the opening marker positions further speeds up the

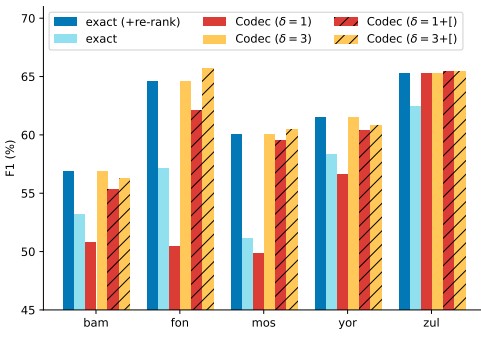

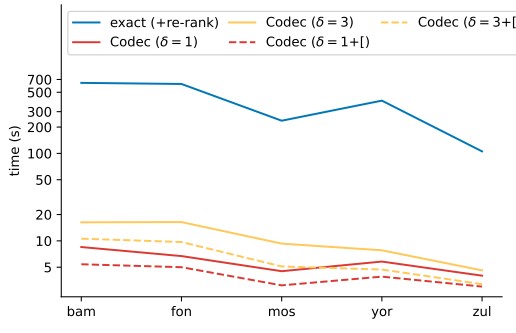

(a) F1 scores on dev set (translate-dev).   (b) Average decoding time per example.

Figure 3: Ablation study on MasakhaNER2.0 dev set of different search settings for five languages, including Bambara (*bam*), Fon (*fon*), Mossi (*mos*), Yoruba (*yor*), and isiZulu (*zul*). **exact (+re-rank)**: exact search with re-ranking; **exact**: exact search and return the top-1 hypothesis. '$\delta$' is the hyperparameter of the heuristic lower bound. '+ [' indicates pruning unlikely opening-marker positions beforehand. Compared to constrained decoding with exact search, CODEC with $\delta = 3$ significantly reduces the decoding time, while retaining the performance measured by F1 scores.

decoding speed by at least about 1.4 times when $\delta = 3$, while causing only a slight drop of F1 (the biggest drop is 0.7 absolute F1 in Yoruba). One interesting observation is that pruning the possible opening-marker position can boost the performance in many cases (i.e., all languages when $\delta = 1$; Fon, Mossi, and isiZulu when $\delta = 3$). One explanation is that the translation model is imperfect and may give bad hypotheses high probabilities, sometimes even higher than the probability of the correct hypothesis. With the marker-position pruning step, those noisy hypotheses are avoided. More analyses about CODEC are in Appendix E.

## 5 MANUAL ASSESSMENT OF CODEC AND DISCUSSION

In this section, we analyze the behavior of each sub-component in CODEC. In particular, we manually inspect outputs of CODEC when being used in translate-train and translate-test settings for the cross-lingual NER task from English to Chinese and Vietnamese (100 examples are inspected for each language). We categorize them into error types based on the source of errors: (1) **Translation**: error from the MT model, (2) **NER**: error from the NER tagger (translate-test only), (3) **CODEC**: error from CODEC. For all studies, we use the translation from Google Translation (GMT) API[2] (which has better translation quality for these two languages) to focus the analysis on CODEC, other than the MT system. In addition, we also inspect the outputs of CODEC in the scenarios when the outputs from the MT model and NER tagger are perfect (i.e., Oracle). More details are provided in the Appendix §F.

**Results** From Table 3, we see that there is a considerable amount of errors from CODEC in Chinese. Further inspecting the error outputs from CODEC, we observe that in more than 60% of the cases, the correct hypothesis is in the top-k found by CODEC, but the re-ranking step fails to retrieve it. Therefore, studying how to select the best hypothesis is one promising direction to improve CODEC further. Also from the table, there are many errors from the MT models in the translate-test setting. In Vietnamese, most of the translation error is incorrectly translating the entities, while in Chinese, most of the time, the error is not due to incorrect translation but the translation style. Particularly, different styles of translation to English may change the entity type, e.g., "Chinese$_{MISC}$ journalists" and "journalists from China$_{LOC}$" can be the translation from the same text, but have different entity types. One possible direction to tackle this issue is to use constrained decoding to control the translation style (e.g., use words that mean nation instead of nationality).

## 6 RELATED WORK

**Label Projection** There has been research that involves projecting label spans between bilingual parallel texts for cross-lingual transfer. One popular approach is to utilize word alignment models to

---
[2]https://cloud.google.com/translate

Table 3: Manual analysis of CODEC outputs in translate-train and translate-test settings for Vietnamese and Chinese. The number indicates how many examples out of the 100 sampled for each language are correct or have corresponding errors (one example may have more than one error).

| Language | MT | Translate-train | | | Translate-test | | | |
|---|---|---|---|---|---|---|---|---|
| | | Correct | Translation | CODEC | Correct | NER | Translation | CODEC |
| Vietnamese | GMT | 94 | 2 | 4 | 51 | 33 | 14 | 13 |
| | Oracle | 96 | 0 | 4 | 85 | 0 | 0 | 15 |
| Chinese | GMT | 80 | 6 | 14 | 34 | 36 | 24 | 16 |
| | Oracle | 83 | 0 | 16 | 69 | 0 | 0 | 31 |

project the label span between the source and target sentence. This approach has been adopted for a wide range of tasks: Part-of-speech Tagging (Yarowsky & Ngai, 2001; Duong et al., 2013; Eskander et al., 2020), Semantic Role Labeling (Akbik et al., 2015; Fei et al., 2020), Named Entity Recognition (Ni et al., 2017; Stengel-Eskin et al., 2019; García-Ferrero et al., 2023; Behzad et al., 2023), and Slot Filling (Xu et al., 2020). With the advancement of the MT systems, several recent efforts have adopted the marker-based approach, which directly uses the MT system to perform the label projection (Lee et al., 2018; Lewis et al., 2020; Hu et al., 2020; Bornea et al., 2021; Chen et al., 2023a). Between the two directions, the alignment-based approach is more adaptable in terms of being applicable to both translate-train and translate-test, and can preserve the translation quality; however, it is shown to be less accurate than the marker-based approach (Chen et al., 2023a). CODEC has the advantages of both approaches. There is also research that takes different directions to transfer label spans, Daza & Frank (2019) presents an encoder-decoder model to translate and generate Semantic Role labels on the target sentence at the same time, while Guo & Roth (2021) proposes to use constrained decoding to construct the target sentence from pre-translated labeled entities. In parallel to our work, Parekh et al. (2023) also propose to translate the original sentence without markers, before label projection, to address the translation quality degradation. In addition to having access to a translation system, their approach requires an instruction-tuned large language model (currently not available in many languages) to detect the label spans in the target language. Different from their method, we adopt an approach based on constrained decoding, which only requires a translation model, such as NLLB, that can support a much wider range of low-resource languages. Outside the topic of label projection, Razumovskaia et al. (2023) share a similar idea of re-purposing multilingual models with ours, and introduces a two-stage framework to adapt multilingual encoders to the task of slot labeling, in the transfer-free scenarios, where no annotated English data is available.

**Lexically constrained decoding** While constrained decoding has previously been explored in machine translation and text generation, it is typically used to constrain the vocabulary or the length of the outputs (Anderson et al., 2017; Hokamp & Liu, 2017; Post & Vilar, 2018; Miao et al., 2019; Zhang et al., 2020; Lu et al., 2021; 2022; Qin et al., 2022). Compared to these past works, CODEC has some important features that make it more suitable for the label projection problem. Firstly, instead of only constraining the occurrence of some words, CODEC constrains the whole decoding sentence to follow a pre-defined template. This feature is important as it allows CODEC to do the projection to a fixed target sentence, which is a requirement for being applicable to the translate-test and preserving translation quality during translate-train. Secondly, CODEC also constrains the number of occurrences of each token, including the marker, which is essential as it prevents the generative model from dropping markers, addressing a serious issue with the marker-based approach for label projection. The idea of using constrained decoding to guarantee the validity of the output space has also been adopted in the task of semantic parsing (Scholak et al., 2021).

## 7 CONCLUSION

In this work, we introduced a new approach of using constrained decoding for cross-lingual label projection. Our new method – CODEC – not only addresses the problem of translation-quality degradation, which is a major issue of the previous marker-based label projection methods, but is also adaptable in terms of being applicable to both translate-train and translate-test. Experiments on two cross-lingual tasks over 20 languages show that, on average, CODEC has shown improvements over strong fine-tuning baseline and other label projection methods.

## 8 ACKNOWLEDGMENTS

We would like to thank Vedaant Shah for helping us with the GPT-4 experiments, and we also thank four anonymous reviewers for their helpful feedback on this work. We would like to thank Azure's Accelerate Foundation Models Research Program for graciously providing access to API-based models, such as GPT-4. This research is supported by the NSF (IIS-2052498) and IARPA via the HIATUS program (2022-22072200004). The views and conclusions contained herein are those of the authors and should not be interpreted as necessarily representing the official policies, either expressed or implied, of NSF, ODNI, IARPA, or the U.S. Government. The U.S. Government is authorized to reproduce and distribute reprints for governmental purposes notwithstanding any copyright annotation therein.

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

## A    CODEC IMPLEMENTATION DETAILS

When we approximate the original $m$-projection problem by solving $m$ 1-projection sub-problems, there are cases where some projections overlap with each other. In translate-train, we remove all examples which have this issue.

### A.1    PRUNING OPENING-MARKER POSITIONS

Let $\mathcal{M}_1$ be a set of all position $i$, in which the difference between the two log probabilities is greater than a threshold $\alpha_1$. Let $\mathcal{M}_2$ be a neighbor set of $\mathcal{M}_1$, which includes all positions that are near each position in $\mathbb{M}_1$ (within a window of size $\sigma$) and have a difference greater than a second threshold $\alpha_2$ ($\alpha_2 < \alpha_1$). The set of candidates for the opening marker position $\mathcal{M}$ is the union of the two sets - $\mathcal{M}_1$ and $\mathcal{M}_2$. In our experiments, we choose $\alpha_1 = 0.5, \alpha_2 = 0.1, \sigma = 5$.

### A.2    SEARCHING FOR TOP-K HYPOTHESES

This step is described in Algorithm 1. In our implementation, for efficiency, we send a batch of partial hypotheses in Line 19. We set the batch size equal to 16 and 12 for NER and EAE experiments, respectively. For all experiments, we search for 5 hypotheses with the highest probabilities (i.e., $k = 5$).

In Eq 8, the max operation is to avoid the case where the current-chosen hypothesis "saves" the markers until the end. In this case, the probability of a prefix of the current-chosen hypothesis is high at the beginning, thus can mistakenly prune the correct hypothesis, but degrades significantly after a marker is inserted. We set $\delta$ equal to 1 for translate-train experiments in the MasakhaNER2.0 dataset and set $\delta$ equal to 5 for all other experiments.

### A.3    RE-RANKING

For the translate-train experiments, we also use an additional lexical-span-level score to filter noisy augmented data. Particularly, we translate each label span in the source sentence independently, let $e^{trans}$ be the translation of a label span $e^{src}$, the lexical-span-level score of the i$^{th}$ hypothesis is the lexical similarity between $e^{trans}$ and $e_i^{tgt}$, where $e_i^{tgt}$ is the span found in the i$^{th}$ hypothesis, following the fuzzy string matching method in Chen et al. (2023a). An example will be filtered out if both its lexical-span-level score and the span-level score defined in §3.4, are smaller than the two corresponding thresholds. We set the threshold for the lexical- and probability-span-level scores as 0.5 and -5 respectively.

| English Data | Augmented data in low-resource languages | | |
| --- | --- | --- | --- |
| | **EasyProject** | **Awesome-align** | **Codec** |
| **India**Loc and **Pakistan**Loc have fought … region of Kashmir Loc … | **India**Loc ne **Pakistan**Loc … ye Kashmir Loc chibviro … | **India**Loc nePakistan … zvinetso yeKashmir Loc … | **India**Loc nePakistan Loc … zvinetso yeKashmir Loc … |
| State media quoted **China**Loc 's top negotiator with **Taipei**Loc , Tang Shubei PER , … from **Taiwan**Loc … | Imithombo … we **China**Loc ne **Taipei**Loc , uTang Shubei PER , … elivela e**Taiwan**Loc … | **Imithombo**Loc … waseChina neTaipei , uTang Shubei PER , … elivela eTaiwan … | Imithombo … **waseChina**Loc ne**Taipei**Loc , uTang Shubei PER , … elivela e**Taiwan**Loc … |

Figure 4: Examples of using different approaches to project label spans from English to low-resource languages (i.e., chiShona (*middle*) and isiZulu (*bottom*)) in the translate-train setting for Cross-lingual NER. In each example, label spans in English data and their corresponding projections in the target language have the same color, the projection errors are underline. In the two examples: (1) **EasyProject** incorrectly splits some words and only marks a part of them as an entity (e.g., "Pakistan" instead of "nePakistan"); (2) **Awes-align** cannot project all label spans and incorrectly map "China" to "Imithombo" in the second example; (3) CODEC has the correct projections in both examples.

---

**Algorithm 1** Constrained_DFS: Searching for top-k best hypotheses

---

**Input** $x^{mark}$: Source sentence with marker, $y$: translation prefix (default: $\epsilon$), $y^{tmpl}$: translation template,
$L$: $[\log P(y_1|x), \log P(y_{1:2}|x), \ldots, \log P(y|x)]$ (default=[0.0]), $\mathcal{M}$: opening marker positions
$H$: min heap to record the results, $k$: number of hypotheses, $\delta$: lower bound hyperparameter

1: $flag \leftarrow \{$check if all markers are generated$\}$
2: **if** $y_{|y|} = $ `` and $flag = $ TRUE: **then**
3:    $H.\text{push}((L_{|y|}, L, y))$                                        ▷ $H$ sorts by the first element
4:    **if** $\text{len}(H) > k$ **then**
5:       $H.\text{pop}()$
6: **else**
7:    $\mathcal{T} \leftarrow []$
8:    $w_1 \leftarrow \{$get the next token in $y^{tmpl}\}$
9:    $\mathcal{T} \leftarrow \mathcal{T} \cup \{(w_1, \log P(w_1|y, x^{mark}))\}$
10:   $j \leftarrow |y| + 1$                                   ▷ position of the token to be generated next
11:   $w_2 \leftarrow \{$get the next marker$\}$
12:   **if** $\exists \, w_2$ and not $(w_2 = $ '[' land $j \notin \mathcal{M})$ **then**
13:      $\mathcal{T} \leftarrow \mathcal{T} \cup \{(w_2, \log P(w_2|y, x^{mark}))\}$
14:   $\mathcal{T} \leftarrow \{$sort $\mathcal{T}$ by the second element in decreasing order$\}$
15:   **for** $(w, p) \in \mathcal{T}$ **do**
16:      $logp \leftarrow L_{|y|} + p$
17:      $\gamma \leftarrow \{$compute lower bound following Eq 7$\}$
18:      **if** $logp > \gamma$ **then**
19:         Constrained_DFS($x^{mark}, y \cdot w, y^{tmpl}, L \cup \{logp\}, \mathcal{M}, H, k, \delta$)
20: **return** $H$

---

## B  EXPERIMENT DETAILS

For all experiments, we use 1 A40 GPUs (48GB each). About fine-tuning backbone models, for the NER model, we fine-tune mDeBERTaV3-base (276M) using the learning rate of 2e-5, batch size 16, and train for 5 epochs (except for the baseline, which is trained for 10 epochs) provided by Chen et al. (2023b). Due to the high variance of cross-lingual transfer performance (Chen & Ritter, 2021), we report average results of 5 random seeds. About the Event Argument Extraction model, for a fair comparison, we fine-tune mT5-large using the X-Gear (Huang et al., 2022) codebase[3], using their provided hyperparameters when fine-tuning and using their scripts for evaluation.

For all experiments, we use the NLLB-3.3B version to generate the translation template and and use NLLB-600M for decoding in CODEC, unless specified otherwise.

## C  MORE EXPERIMENTS ON MASAKHANER2.0

### C.1  MORE BASELINES WITH COMPETITIVE PRE-TRAINED MODELS

In §4, we fine-tune mDebertaV3-base model on English data (FT$_{En}$) as a baseline NER model, and use mBERT-base (Devlin et al., 2019) as the multilingual models in Awesome-align. In this section, we consider using other pre-trained models, which are more competitive for African languages, for the two aforementioned baselines: (1) AfroXLMR-large (Alabi et al., 2022), a model which is adapted to 17 African languages, and (2) Glot500-base (ImaniGooghari et al., 2023), which is pre-trained on 500+ languages. We consider the following baselines: (1) **FT$_{En}$**: fine-tuning mDebertaV3/AfroXLMR/Glot500 on English data and directly evaluate the model, (2) **Awes-align**: using Awesome-align with mBERT/AfroXLMR/Glot500 as the underlined multilingual models, (3) **CODEC**. Since Awes-align and CODEC achieve the best average performance in translate-test (as discussed in §4), in this experiment, we evaluate these two approaches in this setting. For all translate-test experiments, mDebertaV3-base is still used as the English NER model.

Table 4 shows the performance of the above methods on the test set of MasakhaNER2.0 dataset. Replacing mDeBERTaV3 (for FT$_{En}$) and mBERT (for Awes-align) with models that are pre-trained

---

[3]https://github.com/PlusLabNLP/X-Gear/tree/main

Table 4: Cross-lingual NER results of: (1) **FT$_{En}$**: Fine-tuning different pre-trained models (mDeBERTaV3, AfroXLMR-large, and Glot500-base) on English data only, (2) **Awes-align**: Awesome-align with different multilingual models (mBERT, AfroXLMR, and Glot500), (3) **CODEC**: CODEC with two different MT models (M2M-100 and NLLB) for decoding, on the test set of MasakhaNER2.0. CODEC and Awesome-align are evaluated in the translate-test setting. CODEC with NLLB outperforms all other baselines on average ("-": these low-resource languages are not supported by the M2M-100 MT system).

| Languages | FT$_{En}$ | | | Awes-align | | | CODEC | |
|---|---|---|---|---|---|---|---|---|
| | mDeBERTa | AfroXLMR | Glot500 | mBERT | AfroXLMR | Glot500 | NLLB | M2M-100 |
| Bambara | 37.1 | 42.8 | 53.0 | 50.0 | 53.8 | 52.0 | **55.6** | - |
| Ewe | 75.3 | 73.1 | 75.4 | 72.5 | 78.0 | 76.8 | **79.1** | - |
| Fon | 49.6 | 54.6 | 59.4 | 62.8 | **66.0** | 61.2 | 61.4 | - |
| Hausa | 71.7 | **74.8** | 68.0 | 70.0 | 73.7 | 73.1 | 73.7 | 73.1 |
| Igbo | 59.3 | 74.4 | 65.4 | **77.2** | 71.7 | 71.2 | 72.8 | 72.8 |
| Kinyarwanda | 66.4 | 68.4 | 67.0 | 64.9 | 68.2 | 68.1 | **78.0** | - |
| Luganda | 75.3 | 78.9 | 80.9 | 82.4 | 82.1 | **82.5** | 82.3 | - |
| Luo | 35.8 | 40.4 | 42.0 | 52.6 | 51.3 | 51.2 | **52.9** | - |
| Mossi | 45.0 | 45.3 | **55.5** | 48.4 | 49.1 | 48.7 | 50.4 | - |
| Chichewa | 79.5 | **82.2** | 73.8 | 78.0 | 76.7 | 77.3 | 76.8 | - |
| chiShona | 35.2 | 38.4 | 37.6 | 67.0 | **78.6** | 75.6 | 78.4 | - |
| Kiswahili | 87.7 | **88.1** | 84.7 | 80.2 | 80.5 | 79.5 | 81.5 | 82.9 |
| Setswana | 64.8 | 74.4 | 68.8 | **81.4** | 80.7 | 80.5 | 80.3 | - |
| Akan/Twi | 50.1 | 41.9 | 57.9 | 72.6 | 71.7 | **73.5** | 73.5 | - |
| Wolof | 44.2 | 49.0 | 64.5 | 58.1 | 59.0 | 57.1 | **67.2** | - |
| isiXhosa | 24.0 | 26.8 | 27.8 | 52.7 | 67.6 | 63.9 | **69.2** | 67.9 |
| Yoruba | 36.0 | 57.0 | 56.1 | 49.1 | 52.7 | 49.2 | **58.0** | 53.8 |
| isiZulu | 43.9 | 47.3 | 46.5 | 64.1 | 75.5 | 74.9 | **76.9** | 75.7 |
| AVG | 54.5 | 58.8 | 60.2 | 65.8 | 68.7 | 67.6 | **70.4** | - |

or adapted to African languages further improve the average results of both approaches. However, the two approaches still fall behind CODEC with NLLB on average.

## C.2 USING THE M2M MT SYSTEM IN CODEC

In this section, we experiment with M2M-100 (418M parameters) (Fan et al., 2021), another MT system that can support many African languages, in CODEC for constrained decoding. Results are shown in Table 4. CODEC with M2M-100 has similar performance to CODEC with NLLB on most of the languages that it can support, except for isiXhosa, and also outperforms the baseline of fine-tuning on English data with mDeBERTa on 5 over 6 supported languages.

## C.3 IMPACT OF THE SCALE OF THE MT MODEL TO CODEC

There are two places where MT systems are used in CODEC: one for generating the translation template, and the other for decoding. In this section, we explore the impact of the scale of each MT model on the performance of CODEC on the MasakhaNER2.0 dataset, in the translate-test setting.

We first analyze the impact of using different sizes of NLLB (i.e., 600M, 1.3B, and 3.3B) to generate the translation template, and use the same MT system (i.e., NLLB-600M) for constrained decoding. Overall, the average F1 scores of methods using NLLB-600M, NLLB-1.3B and NLLB-3.3B as the template-translator are 68.6, 70.6 and 70.4 respectively (details are in Table 9). We observe the performance significantly improve when changing from NLLB-600M to 1.3B, where NLLB's translation quality also improves the most (Costa-jussà et al., 2022).

Given the same translation templates (i.e., using NLLB-3.3B as the template generator), we explored using CODEC with NLLB-600M, NLLB-1.3B and NLLB-3.3B as the constrained decoding module, and observed that the average performance of the three models are 70.4, 70.1, 70.1 respectively.

Table 5: F1 scores of different methods of two Cross-lingual tasks: Event Trigger Detection (ETD) and Event Argument Extraction (EAE) on ACE-2005 dataset in the translate-train setting.

| Languages | Tasks | $FT_{en}$ | Awes-align | EasyProject | CODEC |
|-----------|-------|-----------|------------|-------------|-------|
| Arabic | ETD | 48.0 | **54.6** | 47.2 | 51.3 |
| | EAE | 26.7 | 33.1 | 26.9 | **33.6** |
| | AVG | 37.4 | **43.9** | 37.1 | 42.5 |
| Chinese | ETD | 52.8 | **54.4** | 51.5 | 53.1 |
| | EAE | 33.5 | 37.1 | 37.2 | **41.2** |
| | AVG | 43.2 | 45.8 | 44.4 | **47.2** |

## D    MORE EXPERIMENTS ON ACE2005 DATASET

### D.1    AN END-TO-END PIPELINE FOR CROSS-LINGUAL EVENT EXTRACTION

In the main paper, following many prior works, we assume to access the gold event trigger at the inference time and only focus on the Event Argument Extraction (EAE) task. In this section, we explore a full pipeline for the EAE task, which includes first using an Event Trigger Detection (ETD) module to detect the event trigger, then using an EAE module to extract the event arguments. To train the ETD model in Arabic/Chinese, we project the event triggers from English to the target language, and fine-tune mDeBERTaV3-base on the concatenation of the English and projected data. We consider the same baselines and follow the same setup as §4.2 for fine-tuning the EAE model. We train each model three times with three different random seeds and report the average F1 scores. We also consider using label projection methods (i.e., Awes-align, CODEC) in the translate-test setting, in which we use English ETD and EAE models to extract the event triggers and event arguments, and use label projection to project the information from English to target languages. Table 5 shows the performance of different methods on the test set of ACE 2005. Overall, Awes-Align (an embedding-based word alignment method) is a very strong baseline for high-resource languages, such as Arabic and Chinese. CODEC is a marker-based method, which is more directly comparable to EasyProject (the state-of-the-art marker-based method). CODEC's strengths are more apparent for low-resource languages and event arguments (i.e., varied nouns, pronouns, noun phrases, and entity mentions) where the multilingual LLM/embeddings are less performant. Event triggers are often the main verbs or action nouns (e.g., "married", "attack"), where word aligner seems to be doing better, especially in high-resource languages.

### D.2    USING THE MBART MT SYSTEM IN CODEC

In this section, we explore using a different MT system for the cross-lingual event extraction task. In particular, we choose mBART50 many-to-many translation system (Tang et al., 2020), which is a competitive open-source MT system for Chinese and Arabic. We use it for generating the translation template (for the three label projection methods) and decoding (for CODEC only). Except of changing the underlining MT systems, we follow the same experiment setup as described in §4.2. We also fine-tune mBART50 many-to-many on the synthetic data which Chen et al. (2023a) provided. The fine-tuned checkpoint in used as the MT system of EasyProject and is used as the decoding MT of CODEC.

Table 6 compares the performance of different label projection methods while using mBART50 MT system. CODEC achieves competitive performance compared to the other label projection baselines. It achieves the best results in Arabic, and is slightly worse than EasyProject in Chinese.

### D.3    TRANSLATE-TEST RESULTS

In this section, we explore the usage of label projection methods in the translate-test setting of the Cross-lingual EAE task. In the setting, label projection methods (i.e., Awesome-align, CODEC) are first used to project event triggers from target languages to English, then an English EAE model is run to extract the arguments, and finally the label projection methods are used to project the

Table 6: F1 scores of different label projection methods using mBART50 MT system on ACE2005. mBART50 is used for both template generation and decoding in CODEC.

| Lang. | $FT_{En}$ | mBART50-many-to-many | | |
|---|---|---|---|---|
| | | Awes-align | EasyProject | CODEC |
| Arabic | 44.8 | 49.1 | 45.4 | **49.5** |
| Chinese | 54.0 | 56.6 | **57.6** | 57.2 |
| AVG | 49.4 | 52.9 | 51.5 | **53.4** |

extracted arguments back to target languages. We experiment with two MT systems for translation template - NLLB-3.3B and Google Machine Translation (GMT). We observe that the tokenizer used in NLLB models lacks some Chinese characters and encodes numerous tokens in the Chinese test sentences as "unknown", therefore, in experiments of CODEC with Chinese data, we use mBART50 many-to-many (Tang et al., 2020) to decode. While in the experiments of CODEC with Arabic data, we continue using NLLB-600M to decode. In §4 and D.2, we use the fine-tuned version of NLLB-600M and mBART50 many-to-many as the decoding MT for CODEC. Specifically, the checkpoints are fine-tuned on the synthetic data from Chen et al. (2023a). The data contains parallel sentences, where the entities on both sides are surrounded by markers. Since the text spans within markers in the synthetic data are only entities, the fine-tuned checkpoints might not be performant when projecting other types of span, such as event triggers, which are often main verbs or action nouns. Consequently, to improve the projection accuracy of CODEC in the EAE task, we explore creating new synthetic data, whose parallel sentences having event triggers and arguments surrounded by markers. This data is then used to fine-tune the decoding MT of CODEC. For constructing the data, we use English Event Trigger Detection (ETD) and EAE models to annotate English sentences. We follow the setup in §D.1 to obtain the two models. The extracted event trigger and arguments are then projected to sentences in target languages by utilizing string matching or alignment-based approaches. For the former method, we use NLLB-3.3B to translate event triggers and arguments, and use string matching to find the corresponding text spans in the target sentences, following Chen et al. (2023a). For the latter approach, we utilize Awesome-align to align the event spans between the source and target sentences. We use the training data of NLLB[4] to obtain parallel sentences, following Chen et al. (2023a). In total, our synthetic data has 10,000 sentence pairs for each language pairs (i.e., English-Chinese, English-Arabic, Chinese-English, Arabic-English).

In Table 7, we report the performance of zero-shot cross-lingual transfer approach (i.e., $FT_{En}$), and translate-test results of label projection methods (i.e., Awes-align and CODEC). We study three versions of CODEC, with the difference comes from the synthetic data used to fine-tune the decoding MT: CODEC: the synthetic data is from Chen et al. (2023a), whose parallel sentences having entity spans surrounded by markers, $CODEC_{match}$ and $CODEC_{align}$: our new synthetic data, whose parallel sentences having event triggers and arguments surrounded by markers, constructed by following the string matching and alignment-based approach, respectively. Overall, we observe that the translation quality has a big impact on the performance of translate-test methods, especially in Chinese. For Chinese, label projection methods can only outperform $FT_{En}$ when switching to use GMT as the template translator. Compared to Awes-align, CODEC and variants have better performance for most of the cases, especially in Arabic. Finally, except for $CODEC_{match}$ in Chinese when using NLLB-3.3B as the template translator, $CODEC_{match}$ and $CODEC_{align}$ significantly outperforms CODEC. The observation demonstrates the benefit of using the synthetic data with event labels surrounded when fine-tuning the decoding MT of CODEC.

# E    COMPARE TO BEAM SEARCH IN THE CONSTRAINED SEARCH SPACE

In this section, we compare CODEC with a modified version of beam search that has the same search space, which we name as CODEC to Constrained-Space Beam Search (CSBS). EasyProject is in fact a method that uses an un-modified version of Beam Search. Similar to CODEC, at each decoding iteration, instead of looking at the whole vocabulary, CSBS considers only the next token from the

---

[4]https://huggingface.co/datasets/allenai/nllb

Table 7: Cross-lingual EAE results of label projection methods using different MT systems in the translate-test setting. "MT" is the MT system used for translation template. In CODEC and variants, mBART50 and NLLB-600M are used for decoding for Chinese and Arabic data, respectively. CODEC$_{match}$/CODEC$_{align}$: the decoding MT in CODEC is fine-tuned on synthetic parallel data, where event arguments and triggers are surrounded by markers (more details are in D.3).

| Languages | MT | FT$_{En}$ | Translate-test | | | |
|---|---|---|---|---|---|---|
| | | | Awes-align | CODEC | CODEC$_{match}$ | CODEC$_{align}$ |
| Arabic | NLLB-3.3B | 44.8 | 41.9 | 42.8 | 45.2 | **47.1** |
| | GMT | | 44.5 | 49.1 | **53.9** | 52.2 |
| Chinese | NLLB-3.3B | 54.0 | 45.8 | 47.8 | 44.6 | **50.4** |
| | GMT | | 56.3 | 54.7 | **56.9** | 56.8 |

Table 8: Cross-lingual NER results of using Constrained-Space Beam Search (CSBS) with different beam sizes and CODEC in the translate-test setting in MasakhaNER2.0. CODEC outperforms CSBS for every language.

| Languages | CSBS | | | | CODEC |
|---|---|---|---|---|---|
| | beam=2 | beam=4 | beam=8 | beam=16 | |
| Bambara | 32.6 | 35.9 | 38.5 | 42.4 | 55.6 |
| Ewe | 54.8 | 61.0 | 66.8 | 70.6 | 79.1 |
| Fon | 21.3 | 24.2 | 29.0 | 32.7 | 61.4 |
| Hausa | 54.4 | 66.4 | 68.7 | 70.1 | 73.7 |
| Igbo | 46.2 | 52.3 | 56.5 | 60.3 | 72.8 |
| Kinyarwanda | 61.5 | 70.7 | 72.7 | 74.7 | 78.0 |
| Luganda | 62.0 | 74.2 | 77.8 | 80.3 | 82.3 |
| Luo | 27.4 | 31.8 | 34.4 | 41.1 | 52.9 |
| Mossi | 22.2 | 25.3 | 29.4 | 34.5 | 50.4 |
| Chichewa | 51.9 | 59.0 | 62.5 | 67.8 | 76.8 |
| chiShona | 68.0 | 74.0 | 75.3 | 76.2 | 78.4 |
| Kiswahili | 57.8 | 73.5 | 75.8 | 77.3 | 81.5 |
| Setswana | 69.2 | 74.7 | 76.7 | 78.1 | 80.3 |
| Akan/Twi | 57.3 | 63.1 | 65.4 | 68.1 | 73.5 |
| Wolof | 39.1 | 45.8 | 50.7 | 54.7 | 67.2 |
| isiXhosa | 49.9 | 59.9 | 64.2 | 67.0 | 69.2 |
| Yoruba | 38.1 | 45.1 | 48.4 | 51.7 | 58.0 |
| isiZulu | 54.2 | 69.4 | 72.2 | 74.6 | 76.9 |
| AVG | 48.2 | 55.9 | 59.2 | 62.3 | 70.4 |

translation template and/or a marker. We compare the two search algorithms in the translate-test setting on the MasakhaNER2.0 dataset. Table 8 illustrates the performance of CSBS with different beam sizes (i.e., 2, 4, 8, 16) and CODEC. As shown in the table, a higher beam size will increase the performance of CSBS, however, even with the beam size of 16, CSBS's average F1 score still falls behind CODEC with a big gap.

## F MANUAL ASSESSMENT SETUP

**Setup** In the translate-train setting, we translate 100 English examples from the CoNLL-2002/2003 multilingual NER datasets (Tjong Kim Sang, 2002) to Vietnamese and Chinese, and use CODEC to project label spans. For the translate-test setting, we inspect 100 Vietnamese examples from the VLSP 2016 NER dataset (Nguyen et al., 2019) and 100 Chinese examples from MSRA-NER dataset (Levow, 2006). When using the NLLB model to encode texts from the MSRA dataset, we observe that NLLB lacks many Chinese tokens. Therefore, for experiments on this dataset, we use a fine-tuned version of the M2M-100 model (Fan et al., 2021) instead. M2M-100 (418M) is fine-tuned on the same synthetic dataset and follows the same script which is used to fine-tune the NLLB models used in CODEC and EasyProject (Chen et al., 2023a).

Table 9: Cross-lingual NER results of CODEC using different scalses of NLLB (600m, 1.3B, and 3.3B) for template translation and for constrained decoding. Results are reported for the translate-test setting of MasakhaNER2.0 dataset ("Translate": the NLLB model used for template translation; "Decode": the NLLB model used for decoding in CODEC).

| Languages | Translate=NLLB-3.3B | | | Decode=NLLB-600M | |
| --- | --- | --- | --- | --- | --- |
| | Decode=NLLB-600M | NLLB-1.3B | NLLB-3.3B | Translate=NLLB-600M | NLLB-1.3B |
| Bambara | 55.6 | 56.0 | 55.8 | 48.4 | 54.9 |
| Ewe | 79.1 | 79.3 | 77.7 | 78.0 | 78.2 |
| Fon | 61.4 | 62.4 | 60.2 | 59.7 | 61.2 |
| Hausa | 73.7 | 73.5 | 73.1 | 72.2 | 73.0 |
| Igbo | 72.8 | 72.9 | 72.8 | 70.5 | 71.3 |
| Kinyarwanda | 78.0 | 71.3 | 71.6 | 76.8 | 78.4 |
| Luganda | 82.3 | 82.4 | 81.7 | 80.9 | 82.5 |
| Luo | 52.9 | 53.1 | 53.1 | 50.2 | 54.7 |
| Mossi | 50.4 | 52.1 | 50.7 | 48.4 | 53.7 |
| Chichewa | 76.8 | 76.7 | 76.9 | 71.5 | 75.8 |
| chiShona | 78.4 | 78.0 | 78.3 | 77.7 | 79.2 |
| Kiswahili | 81.5 | 82.0 | 82.5 | 81.2 | 81.1 |
| Setswana | 80.3 | 81.0 | 81.2 | 78.3 | 79.3 |
| Akan/Twi | 73.5 | 72.7 | 74.6 | 74.9 | 73.9 |
| Wolof | 67.2 | 66.2 | 67.5 | 65.2 | 69.8 |
| isiXhosa | 69.2 | 69.6 | 69.1 | 69.6 | 69.4 |
| Yoruba | 58.0 | 56.5 | 58.7 | 56.2 | 58.6 |
| isiZulu | 76.9 | 76.6 | 76.6 | 74.2 | 75.6 |
| AVG | 70.4 | 70.1 | 70.1 | 68.6 | 70.6 |

