# OpenReview forum: "Constrained Decoding for Cross-lingual Label Projection"
_ICLR.cc/2024/Conference — ICLR 2024 poster_

### Official Review · Reviewer_NyaM · 2023-10-23

**Soundness:** 4 excellent
**Presentation:** 3 good
**Contribution:** 4 excellent
**Rating:** 8
**Confidence:** 4

**Summary:**

This work describes CODEC, a method of generating instances of data in new languages with fine-grained labels transferred from high-resource languages (i.e., English) to low-resource languages (e.g., Bambara). This work intelligently identifies that prior methods such as EasyProject have the drawback of non-natural markers (e.g., BIO tokens) degrading translation quality. To counter this, CODEC instead uses an unconstrained translation as a template and proposes a constrained decoding algorithm to reconcile the template with the annotated input.

This changes the formula of EN to BAM from:
```
EN -> add markers -> MT -> BAM + markers
```
to:
```
EN -> BAM,    EN -> add markers -> [EN+markers, BAM] -> MT w/ constrained decoding -> BAM + markers
```
This removes the issue of MT errors near annotation tokens and provides some reference to check approximate validity during CODEC. This work applies CODEC to both the translate-train and translate-test scenarios of cross-lingual transfer to identify that CODEC has benefits nearly everywhere we can use silver-standard data in cross-lingual transfer. Experiments on NER and event argument extraction identify how CODEC benefits cross-lingual transfer across a wide range of low-resource languages. Ablations and analysis across multiple languages are honest and interpretable in discussing where CODEC is beneficial and does not improve.

**Strengths:**

- This is a very original contribution with wide ranging impact to low-resource cross-lingual transfer. Provided a sufficiently user-friendly codebase, the contributions of CODEC to the field could be widespread. This work also provides a more holistic and thoughtful contribution to the problem than the concurrent https://arxiv.org/abs/2309.08943 . Overall, I think this paper absolutely should be accepted.

- The improvement in both translate-train and translate-test scenarios identify the method as a strong new idea with wide applicabiility. Provided _some_ MT capability, this work helps mitigate the cross-lingual transfer gap to languages with very little study. The work smartly focuses on this scenario (i.e., through MasakhaNER) to support that CODEC works in (approx) the lowest resource scenarios available in modern NLP.

- Succintely describing a constrained decoding method is not easy and this work smartly describes the method visually and mathematically for excellent clarity of the contribution. The frank discussion of complexity and the heuristic approximations for tractability are also honest with tradeoffs discussed in detail to inform future practice.

**Weaknesses:**

- [Minor]: the work could be stronger if this could also be extended to larger models (e.g., >1-5B) and the discussion of applicability on more architectures (enc-only, enc-dec and dec-only) could be more details.

- [Minor]: the paper could also be improved with more comparison against zero-shot results from larger multilingual LLMs. This would be hard on the given 1 48GB GPU setup, but could strengthen the vailidity of the improvement using CODEC. In essence, asking if CODEC works on a larger scale would make the findings more universal.

- [Minor]: it would be enlightening to see CODEC across a benchmark such as XTREME-UP but this likely should be future work not included here.

**Questions:**

- Constrained decoding is also a large topic in semantic parsing and the authors could acknowledge work such as https://arxiv.org/abs/2109.05093

- The sentence "The intuition is that, if we decode the translation template but conditioned on the marker sentence, at the position that needs to be inserted an opening marker, the model would give a high probability to this token, and thus assign a low probability to the token from the template, as illus- trated in Figure 2 (Step 1)." is very long and hard to parse. Consider revising.

- Math format mistake in F2 caption, k -> $k$

- Consider a bulleted list at the end of the intro to make your contributions clearer.

- I think the introduction of Bambara as a language from Africa undersells the low resource importance. Consider a more quantitative phrasing such as "Bambara is a Manding language primarily from Mali with approximately 15 million speakers", using information from WALS and Ethnologue.

---

> ### Author Response · Authors · 2023-11-22
>
> Thank you for your insightful comments. We have edited the paper to fix the typos and acknowledge the work you suggested. We have also updated the paper to cover two of the points you mentioned as below:
>
> > **[Minor]: the work could be stronger if this could also be extended to larger models (e.g., >1-5B) and the discussion of applicability on more architectures (enc-only, enc-dec and dec-only) could be more details.**
>
> Thank you for the suggestion, we have conducted more experiments with different sizes of the MT model used for constrained decoding in Codec and update the results in the Appendix C.3 of the paper.
>
> > **[Minor]: the paper could also be improved with more comparison against zero-shot results from larger multilingual LLMs.**
>
> In our original submission, we fine-tune mDebertaV3-base model on English data (FT_En) as a zero-shot baseline for the Cross-lingual NER task. Inspired by your comments, we have added two multilingual pre-trained models, which are more competitive for African languages, for the zero-shot baseline: AfroXLMR-large [1], a model that is adapted to 17 African languages, and Glot500 [2], a model which is pre-trained on 500+ languages including African languages. Below are results of models on the test set of MasakhaNER2.0. We have updated these results in the Appendix C.1 of the paper.
>
> |  | FT_En | FT_En | FT_En | Codec | Codec |
> |-----------|------------|----------------|---------|-----------------|----------------|
> | **Languages** | mDeBERTaV3 | AfroXLMR| Glot500 | Translate-train | Translate-test |
> | Bambara     | 37.1    | 42.8    | 53.0    | 45.8    | **55.6**    |
> | Ewe         | 75.3    | 73.1    | 75.4    | **79.1**    | **79.1**    |
> | Fon         | 49.6    | 54.6    | 59.4    | **65.5**    | 61.4    |
> | Hausa       | 71.7    | **74.8**    | 68.0    | 72.4    | 73.7    |
> | Igbo        | 59.3    | **74.4**    | 65.4    | 70.9    | 72.8    |
> | Kinyarwanda | 66.4    | 68.4    | 67.0    | 71.2    | **78.0**    |
> | Luganda     | 75.3    | 78.9    | 80.9    | 77.2    | **82.3**    |
> | Luo         | 35.8    | 40.4    | 42.0    | 49.6    | **52.9**    |
> | Mossi       | 45.0      | 45.3    | 55.5    | **55.6**    | 50.4    |
> | Chichewa    | 79.5    | **82.2**    | 73.8    | 76.8    | 76.8    |
> | chiShona    | 35.2    | 38.4    | 37.6    | 72.4    | **78.4**    |
> | Kiswahili   | 87.7    | **88.1**    | 84.7    | 83.1    | 81.5    |
> | Setswana    | 64.8    | 74.4    | 68.8    | 74.7    | **80.3**    |
> | Akan/Twi    | 50.1    | 41.9    | 57.9    | 64.6    | **73.5**    |
> | Wolof       | 44.2    | 49.0    | 64.5    | 63.1    | **67.2**    |
> | isiXhosa    | 24.0      | 26.8    | 27.8    | **70.4**    | 69.2    |
> | Yoruba      | 36.0      | 57.0    | 56.1    | 41.4    | **58.0**    |
> | isiZulu     | 43.9    | 47.3    | 46.5    | 74.8    | **76.9**    |
> | **Average**     | 54.5    | 58.8    | 60.2    | 67.1    | **70.4**    |
>
> [1] Alabi et al. Adapting pre-trained language models to African languages via multilingual adaptive fine-tuning. COLING 2022.
>
> [2] ImaniGooghari et al. “Glot500: Scaling multilingual corpora and language models to 500 languages”. ACL 2023.

---

### Official Review · Reviewer_VywY · 2023-10-26

**Soundness:** 3 good
**Presentation:** 4 excellent
**Contribution:** 3 good
**Rating:** 6
**Confidence:** 3

**Summary:**

This work proposes a constrained-translation-based label projection method for the cross-lingual transfer of two mention extraction problems (named entity recognition and event argument extraction). Instead of directly translating marked sentences, the proposed the method adopts a two-stage approach: first translate the original source without markers, then perform constrained decoding with the marked source and the translation in the first pass. The decoding algorithm consists several interesting parts, including marker-position pruning, branch-and-bound searching and re-ranking. With evaluations on multiple target languages, the proposed method is shown to provide benefits over existing baselines.

**Strengths:**

- The paper is well-written and easy to follow.
- The proposed method is intuitive and effective.

**Weaknesses:**

- The approach relies on an external MT system, whose performance may influence the effectiveness of the label projection. It would be nice if there can be an analysis on the influence of translation quality.
- In some cases, the proposed method does not perform well, for example, the NER results are worse for Chichewa and Kiswahili, and the EAE results seem close to the baselines. It would be much better if there can be more analysis on why these happen to provide some guidance on how to select the label-projection methods for a new language.

**Questions:**

- It would be nice to discuss and measure the efficiency of different methods, especially considering the extra stages of the proposed method. This can be important for “translate-test“, and maybe also for “translate-train” if the cost difference is too much.
- I’m wondering whether it would still be effective to replace the searching algorithm with some simpler alternatives, such as greedy pruning (like in a QA-MRC model). Since the problem itself is inserting a pair of markers, the output space is much smaller than the translation.
- For the event task, it seems that the event triggers are assumed already given? How about considering the full event structures? This might be straight-forward since only span-projection would be enough (it would also be very interesting to consider pairs of spans when projecting).

---

> ### Author Response · Authors · 2023-11-22
>
> Thank you for your insightful feedback. Please find below our response to each point of your comments and some updates in the paper based on your suggestions.
>
> >**The approach relies on an external MT system... It would be nice if there can be an analysis on the influence of translation quality.**
>
> Thank you for your suggestion, we have conducted additional analysis about this topic and updated the paper with the new analysis in Appendix C.3. The update about this analysis is as follows. There are two places where MT systems are used in Codec: one is for generating the translation template, and the other one is for decoding.
> * We analyzed the impact of using different sizes of NLLB (600M, 1.3B, and 3.3B) to generate the translation templates, while fixed on NLLB-600M as the MT system for constrained decoding. We evaluate the methods in the translate-test setting on the MasakhaNER2.0 dataset, over 18 languages. Overall, the average F1 scores of methods using NLLB-600M, NLLB-1.3B and NLLB-3.3B as the template-translator are 68.6, 70.6 and 70.4 respectively. We observe the performance improves the most when changing from NLLB-600M to 1.3B, where NLLB’s translation quality also improves the most [1].
> * Given the same translation templates (i.e., using NLLB-3.3B as the template-generator), we explored using Codec with the three model sizes of NLLB as the constrained decoding module. The average F1 score of methods using NLLB-600M, NLLB-1.3B, and NLLB-3.3B are 70.4, 70.1, and 70.1, respectively.
>
> [1] NLLB Team. No Language Left Behind: Scaling Human-Centered Machine Translation. https://arxiv.org/abs/2207.04672
>
> >**In some cases, the proposed method does not perform well, for example, the NER results are worse for Chichewa and Kiswahili, and the EAE results seem close to the baselines. It would be much better if there can be more analysis on why these happen to provide some guidance on how to select the label-projection methods for a new language.**
>
> For the case of Chichewa and Kiswahili, we observe that, for a large portion of the data, the entities in the target language are exactly the same as their English form, therefore, the multilingual model fine-tuned only on English data does extremely well in these two languages , and outperforms all label projection methods. We have added a discussion about this in the Experiments section.
> >**I’m wondering whether it would still be effective to replace the searching algorithm with some simpler alternatives, such as greedy pruning...**
>
> We have implemented a simpler search algorithm, we dub it as Constrained-Space Beam Search (CSBS). In particular, this algorithm behaves similarly to the vanilla beam search, but we constrain the search space of it to contain only valid hypotheses - following the translation template and having the correct number of markers.  Similar to Codec, CSBS also returns up to 5 best hypotheses (equal to the beam size when the beam size is smaller than 5), and a re-ranking step is performed to choose the best hypothesis. Below is the performance of CSBS with different beam sizes and Codec on the MasakhaNER2.0 test set. The methods are evaluated in the translate-test setting (see below). We have updated these results in the Appendix E of the paper. Overall, a higher beam size will increase the performance of CSBS, however, even with a beam size of 16, CSBS’s average F1 score still falls behind Codec with a big gap.
>
> |  | CSBS | CSBS | CSBS | CSBS | Codec |
> |-------------|--------|--------|--------|---------|-------|
> | **Languages**   | beam=2 | beam=4 | beam=8 | beam=16 |  |
> | Bambara     | 32.6   | 35.9   | 38.5   | 42.4    | 55.6  |
> | Ewe         | 54.8   | 61.0   | 66.8   | 70.6    | 79.1  |
> | Fon         | 21.3   | 24.2   | 29.0   | 32.7    | 61.4  |
> | Hausa       | 54.4   | 66.4   | 68.7   | 70.1    | 73.7  |
> | Igbo        | 46.2   | 52.3   | 56.5   | 60.3    | 72.8  |
> | Kinyarwanda | 61.5   | 70.7   | 72.7   | 74.7    | 78.0  |
> | Luganda     | 62.0   | 74.2   | 77.8   | 80.3    | 82.3  |
> | Luo         | 27.4   | 31.8   | 34.4   | 41.1    | 52.9  |
> | Mossi       | 22.2   | 25.3   | 29.4   | 34.5    | 50.4  |
> | Chichewa    | 51.9   | 59.0   | 62.5   | 67.8    | 76.8  |
> | chiShona    | 68.0   | 74.0   | 75.3   | 76.2    | 78.4  |
> | Kiswahili   | 57.8   | 73.5   | 75.8   | 77.3    | 81.5  |
> | Setswana    | 69.2   | 74.7   | 76.7   | 78.1    | 80.3  |
> | Akan/Twi    | 57.3   | 63.1   | 65.4   | 68.1    | 73.5  |
> | Wolof       | 39.1   | 45.8   | 50.7   | 54.7    | 67.2  |
> | isiXhosa    | 49.9   | 59.9   | 64.2   | 67.0    | 69.2  |
> | Yoruba      | 38.1   | 45.1   | 48.4   | 51.7    | 58.0  |
> | isiZulu     | 54.2   | 69.4   | 72.2   | 74.6    | 76.9  |
> | **Average**         | 48.2   | 55.9   | 59.2   | 62.3    | **70.4**  |

---

> > ### Author Response · Authors · 2023-11-22
> >
> > >**It would be nice to discuss and measure the efficiency of different methods, especially considering the extra stages of the proposed method...**
> >
> > EasyProject is the fastest in the “translate-train” setup, as it does not require any additional step as Codec or Awesome-align; however, EasyProject is not applicable in the “translate-test” setup. Both Awesome-align and Codec can operate under the “translate-test” setup, which will entirely skip the training (fine-tuning) step of NER/EAE models on each target language as required by “translate-train”. We also provided a run time analysis for Codec in Figure 3b in the paper.
> >
> > >**For the event task, it seems that the event triggers are assumed already given? How about considering the full event structures?...**
> >
> > In our original submission, following many prior works, we used gold event triggers for the Event Argument Extraction (EAE) task, in order to do a more controlled experiment where variance is reduced due to errors in the first step of the pipeline for end-to-end event detection.
> >
> > We added a new experiment, in which an Event Trigger Detection model is first used to detect the event triggers; then given the input sentence and the detected triggers, the Event Argument Extraction model extracts the event arguments. For the Event Trigger Detection model, we fine-tune mDeBERTaV3-base on the concatenation of English data, and projected event trigger data in Arabic/Chinese. Details about the experiments and results have been updated in Appendix D.1 of the paper. Below are the results of different methods in this pipeline setting:
> >
> > | Languages | Tasks | FT_En | Awes-align | EasyProject | Codec |
> > |-----------|-------|-------|------------|-------------|-------|
> > | Arabic    | ETD    | 48.0  | **54.6**       | 47.2        | 51.3  |
> > |           | EAE   | 26.7  | 33.1       | 26.9        | **33.6**  |
> > |           | Average   | 37.4  | **43.9**       | 37.1        | 42.5  |
> > | Chinese   | ETD    | 52.8  | **54.4**       | 51.5        | 53.1  |
> > |           | EAE   | 33.5  | 37.1       | 37.2        | **41.2**  |
> > |           | Average  | 43.2  | 45.8       | 44.4        | **47.2**  |
> >
> > ("ETD": Event Trigger Detection, "EAE": Event Argument Extraction, "FT_En": Fine-tune the model on English data only)
> >
> > Overall, Awesome-Align (an embedding-based word alignment method) is a very strong baseline for high-resource languages, such as Arabic and Chinese. Codec is a marker-based method, which is more directly comparable to EasyProject (the state-of-the-art marker-based method). Codec's strengths are more apparent for low-resource languages and event arguments (i.e., varied nouns, pronouns, noun phrases, and entity mentions) where the multilingual LLM/embeddings are less performant. Event triggers are often the main verbs or action nouns (e.g., “married”, “attack”), where word aligner seems to be doing better, especially in high-resource languages. The results are a little bit more mixed, as the errors can happen in either trigger detection step (then propagate) or the event argument step, or both, which cause more variances (i.e., a less controlled experiment than when gold triggers are given).
> >
> > Thanks again for your comments!

---

### Official Review · Reviewer_kHv7 · 2023-10-31

**Soundness:** 2 fair
**Presentation:** 3 good
**Contribution:** 2 fair
**Rating:** 6
**Confidence:** 4

**Summary:**

This paper focuses on improving label projection for zero-shot cross-lingual transfer learning. They claim that existing label projection techniques cannot generate accurate translation and there for affect the downstream performance. They accordingly propose a constrained decoding to decide which positions to insert markers conditioned on a better translation template. Some heuristic tricks are presented to accelerate the search process. Experiments on NER and EAE show the potential of the proposed method.


==== After response ====
Given that the authors promise that they will make the description clearer and add the translate-test results for EAE, I consider increasing the score.

**Strengths:**

- The writing is clear.
- The proposed method performs well on two tasks.

**Weaknesses:**

- The author say that inserting markers would degrade the translation quality. However, although they provide a translation template to guide the model during translation, the proposed method still relies on markers, which is not completely solving this issue.
- When searching, they mention the assumption:
```
If we decode the translation template but conditioned on the marker sentence, at the position that needs to be inserted an opening marker, the model would give a high probability to this token, and thus assign a low probability to the token from the template.
```
This assumption largely relies the translation model’s ability to handle markers. Different translators may have different behaviors to handle the markers. I suggest the authors to report the results of different translators to show the stability of the proposed method.
- The proposed method is based on some heuristic. It would be great if the authors can provide some theoretical bound to justify the heuristic.
- It seems like that they follow the experimental setting of previous work (Chen et al. 2023a). However, they consider MasakhaNER 2.0 rather than WikiAnn (reported by Chen et al. 2023a) for NER without any explanation. Is it because that the proposed method works better for low-resource languages? I suggest to report the scores on WikiAnn as well for more comprehensive comparisons.
- I am a little bit confused by the reason for not considering translation-test for EAE. The authors mention that the English gold trigger is needed but not feasible. However, this can be obtained by applying the proposed constraint label project from the target language to English. It would be interesting to study more about this.

**Questions:**

Please see above.

---

> ### Author Response · Authors · 2023-11-22
>
> Thank you for your insightful feedback. Please find below our response to each point of your comments and some updates in the paper based on your suggestions.
>
> > **The author say that inserting markers would degrade the translation quality. However, although they provide a translation template to guide the model during translation, the proposed method still relies on markers, which is not completely solving this issue.**”
>
> There seems to be some confusion here, our proposed method, Codec, does not include markers during the translation phase; so, the final translated text quality from Codec will be the same as any MT system, which is also the same as the word alignment-based label projection approaches. Only after this translation phase (i.e., after we generated this translation template), the markers are inserted through a second constrained decoding phase, in which the translation is constrained to be the same as the original translation template, while markers are being inserted. We have updated the Introduction of the paper to make this point clearer.
>
> >**This assumption largely relies the translation model’s ability to handle markers... I suggest the authors to report the results of different translators to show the stability of the proposed method.**”
>
> Thank you for your suggestion. We have added experiments with two more different MT systems (M2M-100 and MBart), besides NLLB.
>
> * For the NER task, we explore using M2M-100 instead of NLLB for decoding. M2M-100 is chosen for NER as it supports 6 African languages in the MasakhaNER2.0 dataset. In addition, for this task, we also experiment with different model sizes of NLLB (i.e., 600M, 1.3B, and 3.3B) as the decoding module for Codec. See the table below (under “Codec” column) for results under the translate-test setting.  We have also included the results of these analyses in Appendix C of the paper.
>
> |     | FT_En | CSBS | CSBS | CSBS | CSBS | Codec | Codec | Codec | Codec |
> |--------------|-------|--------|--------|--------|---------|-----------|-----------|---------|--------|
> | Languages    |  | beam=2 | beam=4 | beam=8 | beam=16 | NLLB-600M | NLLB-1.3B | NLLB-3B | M2M-100 |
> | Bambara      | 37.1  | 32.6   | 35.9   | 38.5   | 42.4    | 55.6      | **56.0**      | 55.8    | -      |
> | Ewe          | 75.3  | 54.8   | 61.0     | 66.8   | 70.6    | 79.1      | **79.3**      | 77.7    | -      |
> | Fon          | 49.6  | 21.3   | 24.2   | 29.0   | 32.7    | 61.4      | **62.4**      | 60.2    | -      |
> | Hausa        | 71.7  | 54.4   | 66.4   | 68.7   | 70.1    | **73.7**      | 73.5      | 73.1    | 73.1   |
> | Igbo         | 59.3  | 46.2   | 52.3   | 56.5   | 60.3    | 72.8      | **72.9**      | 72.8    | 72.8   |
> | Kinyarwanda  | 66.4  | 61.5   | 70.7   | 72.7   | 74.7    | **78.0**      | 71.3      | 71.6    | -      |
> | Luganda      | 75.3  | 62.0     | 74.2   | 77.8   | 80.3    | 82.3      | **82.4**      | 81.7    | -      |
> | Luo          | 35.8  | 27.4   | 31.8   | 34.4   | 41.1    | 52.9      | **53.1**      | **53.1**    | -      |
> | Mossi        | 45.0  | 22.2   | 25.3   | 29.4   | 34.5    | 50.4      | **52.1**      | 50.7    | -      |
> | Chichewa     | 79.5  | 51.9   | 59.0   | 62.5   | 67.8    | 76.8      | 76.7      | **76.9**    | -      |
> | chiShona     | 35.2  | 68.0     | 74.0   | 75.3   | 76.2    | **78.4**      | 78.0      | 78.3    | -      |
> | Kiswahili    | 87.7  | 57.8   | 73.5   | 75.8   | 77.3    | 81.5      | 82.0      | 82.5    | **82.9**   |
> | Setswana     | 64.8  | 69.2   | 74.7   | 76.7   | 78.1    | 80.3      | 81.0      | **81.2**    | -      |
> | Akan/Twi     | 50.1  | 57.3   | 63.1   | 65.4   | 68.1    | 73.5      | 72.7      | **74.6**    | -      |
> | Wolof        | 44.2  | 39.1   | 45.8   | 50.7   | 54.7    | 67.2      | 66.2      | **67.5**    | -      |
> | isiXhosa     | 24.0    | 49.9   | 59.9   | 64.2   | 67.0    | 69.2      | **69.6**      | 69.1    | 67.9   |
> | Yoruba       | 36.0    | 38.1   | 45.1   | 48.4   | 51.7    | 58.0      | 56.5      | **58.7**    | 53.8   |
> | isiZulu      | 43.9  | 54.2   | 69.4   | 72.2   | 74.6    | **76.9**      | 76.6      | 76.6    | 75.7   |
> | **Average**          | 54.5  | 48.2   | 55.9   | 59.2   | 62.3    | **70.4**      | 70.1      | 70.1    |    -    |
>
> (‘-’: these low-resource languages are not supported by the M2M-100 MT system)
>
> * For the EAE task, we have conducted additional experiments using MBart50-large (a competitive MT system for Chinese and Arabic) for both translation template generation and constrained decoding in Codec, as well as other comparison baselines. See table below for results. We have updated the paper (Appendix D.2) to include these results.
>
>  Languages   | FT_En | Awes-align | EasyProject | Codec |
> |---|---|---|---|---|
> | Arabic  | 44.8  | 49.1  | 45.4  | **49.5** |
> | Chinese | 54.0  | 56.6 | **57.6** | 57.2 |
> | **Average**     | 49.4  | 52.9 | 51.5 | **53.4** |

---

> > ### Author Response · Authors · 2023-11-22
> >
> > > **The proposed method is based on some heuristic. It would be great if the authors can provide some theoretical bound to justify the heuristic.**
> >
> > Similar to many papers on NLP research, we acknowledge that we focus on empirical research, design and experiments for new methods. We agree that theoretical analysis of our new approach to constrained decoding based on branch-and-bound search would be valuable, however we view this as beyond the scope of this paper. That said, inspired by your comment, we have added an experiment to compare Codec to a variant of Beam Search, we dub it as Constrained-Space Beam Search (CSBS). This algorithm behaves similarly to the vanilla beam search, but we constrain the search space of it to contain only valid hypotheses -- following the translation template and having the correct number of markers. We have reported the performance of CSBS with different beam sizes and Codec in the translate-test setting of the MasakhaNER2.0 (please refer to the table in our reply to your second concern). In summary, higher beam size will increase the performance of CSBS, however, even with the beam size of 16, CSBS’s average F1 score still falls behind Codec with a big gap. We have updated these results in the Appendix E of the paper.
> >
> >
> > > **It seems like that they follow the experimental setting of previous work (Chen et al. 2023a). However, they consider MasakhaNER 2.0 rather than WikiAnn (reported by Chen et al. 2023a) for NER without any explanation...**
> >
> > Thank you for pointing this out, we excluded the older WikiAnn dataset (from 2019) because, as pointed out by the paper [1], there are concerns about the noise level in this automatically constructed dataset. MasakhaNER 2.0 is the newer, higher-quality multilingual NER dataset, which is fully manually annotated. MasakhaNER is increasingly being used in benchmarking since its release in late 2022. We have added an explanation about the exclusion of WikiAnn in section 4.1 of our paper.
> >
> > [1] Lignos et al. Toward more meaningful resources for lower-resourced languages. In Findings of ACL 2022.
> >
> > >**I am a little bit confused by the reason for not considering translation-test for EAE. The authors mention that the English gold trigger is needed but not feasible...**
> >
> > We agree that the English trigger can be achieved by using another label projection from the target language to English. We have updated the Experiment section in the paper to make this point clearer.
> >
> > Thanks again for all your comments and suggestions, which helped us improve the paper!

---

> > ### Comment · Reviewer_kHv7 · 2023-11-22
> >
> > Thanks for your response. Let me explain my thought about the first comment a little bit more.
> >
> > I understand that Codec does not include markers during the translation phase. However, what I want to say is that. For prior marker-based approaches, based on my understanding, their drawback can be summarized as
> >
> > input with markers -> *output word probability* becomes bad (due to markers) -> translated sentence is bad
> >
> > I think adding markers makes the output word probability abnormal for certain degree and this is the key point why the translation quality degrades. For Codec, although it has a translated sentence being independent to the markers, the marker insertion still replies on *output word probability* (which is based on the input with markers) to decide which position we should insert the markers. That's why I say the proposed method still relies on markers. I agree that Codec uses a smarter way to generate markers, but it does not solve the problem that *output word probability* becomes inaccurate because of adding markers. Given this, I feel the paper is a bit over-claiming.
> >
> > Any discussion is welcome. Thanks!

---

> > > ### Comment · Reviewer_kHv7 · 2023-11-22
> > >
> > > Also, it would be great if you can report the translation-test results for EAE as well, since you emphasize the strength of Codec on translation-test in the introduction.

---

> > > > ### Author Response · Authors · 2023-11-22
> > > >
> > > > > **Thanks for your response. Let me explain my thought about the first comment a little bit more....**
> > > >
> > > > Thank you for clarifying! Now we understand where the confusion is. There are two types of issues: projection accuracy and translation quality.
> > > >
> > > > In terms of the projection accuracy, we agree that the output word probability might become inaccurate because of adding markers and the MT model might have trouble determining where to insert markers, but we do not claim to completely solve this issue. In our work, we try to improve the projection accuracy of Codec and empirically prove that our approach works better than the alignment- and marker-based baselines. However, there is still room for further improvement regarding this problem.
> > > >
> > > > About the translation quality, this is our main point about the benefit of Codec over previous marker-based approaches. In particular, the translation template, which Codec uses to insert markers, has higher quality than the translation from prior marker-based approaches.
> > > >
> > > > Thank you again for initiating this discussion and we can make it clear to resolve the confusion. We will update the camera-ready version of the paper to make sure this point is clear.
> > > >
> > > > >**Also, it would be great if you can report the translation-test results for EAE as well, since you emphasize the strength of Codec on translation-test in the introduction.**
> > > >
> > > > Thank you for the suggestion. We will make sure to add this experiment in the camera-ready, however, given the limited time left for author response, we do not have enough time to finish this experiment. We have provided a lot of additional experiments in response to the review comments.

---

### Official Review · Reviewer_TbXV · 2023-11-02

**Soundness:** 3 good
**Presentation:** 4 excellent
**Contribution:** 3 good
**Rating:** 8
**Confidence:** 4

**Summary:**

=== AFTER THE RESPONSE ===
I would like to thank the authors for taking the time to provide a very detailed response which clarified my main concerns and extra questions. I still think that the method could have been evaluated on a larger selection of tasks, but this doesn't invalidate the soundness of the proposed methodology, and I'm happy to increase my score
===========================

This paper targets cross-lingual transfer for sequence labeling task, where the main problem in previous work has been projecting labeled spans from the source language to the correct spans in the target language, a problem sometimes referred to as the labeled span mismatch. Previous work typically solved this problem via two different approaches: 1) using external word aligners to do the label projection from source to target, or 2) inserting marker tokens directly into the input of a strong (N)MT system and basically conducting a standard translate-train approach (but with those extra markers). However, both prior approaches have issues as the former critically relies on the quality of the external word aligner, while the latter yields to degraded MT performance (due to the insertion of markers).

This paper basically proposes an extension to the latter approach, aiming to preserve the original quality of the MT system by bypassing the direct insertion of markers, and proposes a two-step approach where in Step 1) the original text can be translated (via translate-train or translate-test), and in Step 2) projection is added via constrained decoding, keeping the translation from Step 1 as a fixed template. The main technical contribution is then a computationally feasible technique for the constrained decoding, bypassing the need to conduct exhaustive brute force search while maintaining strong performance. The separation of the translation and marker insertion steps also allows the approach to be applied to the translate-test setting, and the results confirm the usefulness of the technique.

**Strengths:**

- The paper clearly defines the problem, which is very concrete, and sets out to solve the problem following a clear line of thinking: from the conceptual level all the way to low-level technical execution aiming to improve the performance-versus-efficiency trade-off.
- The idea of constrained decoding which fixes the entire sentence (instead of focusing only on lexical constraints during constraints) is quite new (at least to the best of my knowledge) and could have applications beyond cross-lingual transfer tasks discussed in this work.
- The paper is well written and it is easy to link the main hypotheses to the concrete experiments and analyses. The core section of the paper on "constrained decoding" is also nicely described and easy to follow.
- The results on the two tasks seem to support the main claims although the paper requires more experiments (see also under weaknesses).

**Weaknesses:**

- The main issue with the work is 1) the lack of recognition of other (recent and less recent) work on the same problem of cross-lingual label projection, which consequently leads to the 2) lack of more comprehensive comparisons to more baselines. The main baseline is definitely the EasyProject method and I agree with that, but I feel that not enough care has been provided to optimise the word alignment-based baselines which also shows reasonable performance, and is quite competitive in the EAE task.
-- For instance, there has been some recent work on alignment correction for label projection (https://aclanthology.org/2023.law-1.24.pdf), and there are also other very relevant papers which should be cited and discussed (and ideally even compared against): https://aclanthology.org/2021.findings-acl.396.pdf or https://d-nb.info/1203127499/34,
-- The number of evaluation tasks is slightly underwhelming and the paper should extend the scope of tasks to other sequence labeling tasks (e.g., slot labeling in dialogue, dependeny parsing or semantic role labeling) - NER with only 3 NE classes is a (relatively) simple task (from the perspective of its experimental setup), and the paper would have more impact with a wider experimental setup.
-- I would also like to see a wider exploration of different MT systems and chosen encoder-decoder models and their impact on the performance of both alignment-based approaches as well as EasyProject and CODEC. For instance, how would larger variants of NLLB affect the performance? Would the scale of the NMT system recover for its deficiencies?

**Questions:**

A similar two-step idea, but which is not MT-based but encoder-based has been investigated here: https://arxiv.org/pdf/2305.13528.pdf (the idea there is slightly different and is based on classification - in the first step, the system just decides whether something should be a labeled span or not; in the second step, the actual label is added to each span detected as 'labeled span'. The paper should also discuss ideas like this one in related work and they seem highly relevant.

---

> ### Author Response · Authors · 2023-11-22
>
> Thank you for your insightful feedback. Please find below our response to each point of your comments and some updates in the paper based on your suggestions.
>
> In response to your two main concerns:
>
>
> > **The main issue with the work is 1) the lack of recognition of others ... 2) lack of more comprehensive comparisons to more baselines. The main baseline is definitely the EasyProject method and I agree with that, but I feel that not enough care has been provided to optimize the word alignment-based baselines ...**
>
> For 1), thank you for pointing this out. There is indeed a long history of research on cross-lingual label projection. We should cite as many papers as possible as we can fit in the page limit. We have updated our Related Work section to include all the papers you mentioned.
>
> For 2), thank you for your suggestion. Our paper focuses on improving the marker-based approach for label projection (especially low-resource languages), thus EasyProject, the best marker-based approach is used as the most important baseline for comparison as you mentioned. In terms of comparison to the word alignment-based approaches, in our original submission, we followed the prior work [1] in choosing Awesome-align (one of the state-of-the-art word alignment models) with multilingual BERT as the baseline. In response to your comments, we have added two more word-alignment baselines that are optimized for African languages, namely Awesome-align with AfroXLMR-large [2], a model that is adapted to 17 African languages, and Awesome-align with Glot500 [3], a model which is pre-trained on 500+ languages including African languages. These are novel and very competitive word alignment-based baselines for the MasakhaNER2.0 dataset.
>
> The table below compares the fine-tuning of English data baseline (FT_En), and Awesome-align with multilingual BERT (result in the main paper), Awesome-align with two aforementioned pre-trained models, and our proposed Codec in the translate-test setting, the results are reported on the test set of MasakhaNER2.0 dataset. To save space, in this table, we also include the experiments of using Codec with different MT models (NLLB and M2M-100) for decoding, which is to respond to your later comments. We have updated the paper (Appendix C.1) to include these results.
>
> | | FT_En | Awes-align | Awes-align | Awes-align | Codec | Codec|
> --- | --- | --- | --- | --- | ---|---|
> **Languages** || mBERT | AfroXLMR | Glot500 | NLLB | M2M-100
> Bambara | 37.1 | 50.0 | 53.8 | 52.0 | **55.6** | -
> Ewe | 75.3 | 72.5 | 78.0 | 76.8 | **79.1** | -
> Fon | 49.6 | 62.8 | **66.0** | 61.2 | 61.4 | -
> Hausa | 71.7 | 70.0 | **73.7** | 73.1 | **73.7** | 73.1
> Igbo | 59.3 | **77.2** | 71.7 | 71.2 | 72.8 | 72.8
> Kinyarwanda | 66.4 | 64.9 | 68.2 | 68.1 | **78.0** | -
> Luganda | 75.3 | 82.4 | 82.1 | **82.5** | 82.3 | -
> Luo | 35.8 | 52.6 | 51.3 | 51.2 | **52.9** | -
> Mossi | 45.0 | 48.4 | 49.1 | 48.7 | **50.4** | -
> Chichewa | **79.5** | 78.0 | 76.7 | 77.3 | 76.8 | -
> chiShona | 35.2 | 67.0 | **78.6** | 75.6 | 78.4 | -
> Kiswahili | **87.7** | 80.2 | 80.5 | 79.5 | 81.5 | 82.9
> Setswana | 64.8 | **81.4** | 80.7 | 80.5 | 80.3 | -
> Akan/Twi | 50.1 | 72.6 | 71.7 | **73.5** | **73.5** | -
> Wolof | 44.2 | 58.1 | 59.0 | 57.1 | **67.2** | -
> isiXhosa | 24.0 | 52.7 | 67.6 | 63.9 | **69.2** | 67.9
> Yoruba | 36.0 | 49.1 | 52.7 | 49.2 | **58.0** | 53.8
> isiZulu | 43.9 | 64.1 | 75.5 | 74.9 | **76.9** | 75.7
> **Average** | 54.5 | 65.8 | 68.7 | 67.6 | **70.4**| -
>
> (‘-’ marks the low-resource languages that are not supported by the M2M MT system)
>
> [1] Chen et al. Frustratingly easy label projection for cross-lingual transfer. Findings ACL 2023.
>
> [2] Alabi et al. Adapting pre-trained language models to African languages via multilingual adaptive fine-tuning. COLING 2022.
>
> [3] ImaniGooghari et al. “Glot500: Scaling multilingual corpora and language models to 500 languages”. ACL 2023.
>
> In response to your other comments:
>
> > **The number of evaluation tasks is slightly underwhelming and the paper should extend the scope of tasks ...**
>
> Our proposed work primarily targets low-resource languages and span-level NLP tasks, where human-annotated datasets (that are directly annotated on these languages, other than annotating the translated texts from existing English datasets) are in scarcity. MasakhaNER2.0, which covers 18 African languages we can experiment on, is one of the best benchmarks that is available. Besides the NER task, we also did experiments on the ACE 2005 dataset for the EAE task, which is a commonly used benchmark in the label projection literature. We politely argue that having conducted experiments in 20 languages and two tasks (with 3 different MT systems, and 4 competitive baselines -- see our updates) are sufficient to support our claimed contributions, while agreeing with you that it is always nice to include even more experiments if possible.

---

> > ### Author Response · Authors · 2023-11-22
> >
> > > **I would also like to see a wider exploration of different MT systems...**
> >
> > - We have added experiments with two more open-sourced MT systems. For the cross-lingual NER task, we use M2M-100 in place of NLLB as the MT model for decoding, and report the performance of this approach in the translate-test setting, on the MasakhaNER2.0 dataset. Please refer to the table at the beginning of our response for the added results. In summary, Codec using M2M-100 as the decoding module has similar performance Codec using NLLB on most of the languages, except for isiXhosa, and also outperforms the baseline of fine-tuning on English data (FT_En) on 5 over 6 supported languages. We have updated these results in the Appendix C.2 of the paper.
> >
> >     For the Event Argument Extraction task, we have also conducted additional experiments of using MBart50-large (a competitive MT system for Chinese and Arabic) for both translation template generation and constrained decoding in Codec, as well as comparison baselines. See table below. We have updated the paper (Appendix D.2) to include these results.
> >
> > | Lang.   | FT_En | Awes-align | EasyProject | Codec |
> > |---------|-------|----------------------|------------|-------------|
> > | Arabic  | 44.8  | 49.1                 | 45.4       | **49.5** |
> > | Chinese | 54.0  | 56.6                 | **57.6**       | 57.2 |
> > | Average     | 49.4  | 52.9                 | 51.5       | **53.4** |
> >
> > - In terms of the scales of NMT systems, we want to clarify that, in our original submission, both the Awesome-align and EasyProject baselines were using one of the larger NMT models (NLLB-3.3B). In response to your comment, we further added some experiments (each number is averaged from 18 languages) with different scales of NMT models for Codec on MasakhaNER2.0 in the Translation-test setup:
> >
> >     - Different scales of MT for template generation in Codec (when the decoder is fixed to NLLB-600M): the average F1 scores of NLLB-600M, NLLB-1.3B, and NLLB-3.3B are 68.6, 70.6, and 70.4, respectively.
> >
> >     - Different scales of MT for decoding in Codec (when the template is fixed to that generated by NLLB-3.3B): the average F1 scores of NLLB-600M, NLLB-1.3B, and NLLB-3.3B are 70.4, 70.1, 70.1, respectively.
> >
> >     We have included the above results in Appendix C.3 of the paper.
> >
> > We appreciate all your suggestions. We updated the Related Work section and added multiple experiments that compared with more word alignment baselines, different MT systems, and varied model scales. If you found these additional experiments have helped to address your comments, we would appreciate it if you could consider increasing the review scores and/or acknowledge that.

---

> > > ### Comment · Reviewer_TbXV · 2023-11-23
> > >
> > > Many thanks for providing a detailed response which helped me clarify some main concerns, and in turn this reflects in an increased recommendation score. While the paper might be more interesting for the NLP audience (given the set of chosen evaluations and the fact that it's based on MT systems), it might be interesting also for the wider ML/ICLR audience.

---

### Meta-Review · Area_Chair_U17A · 2023-12-07

**Metareview:**

This paper presents another cross-lingual annotation projection method. There are two differences from existing more common approaches: first, the target level sentence is actually generated by translation and doesn't exist prior to that; second, the tags are not being projected during translation but post-translation. The method first translates the sentence, and then using a constrained form of decoding that is computationally feasible projects the tags from the source sentence to the target sentence. The experiments and the methodlogy is sound and the experiments are thorough. Except, I do feel that in the new era of LLMs, there should be a simple baseline where an LLM is prompted to do this task with few-shot examples, and then that should act as a baseline. I wouldn't be surprised if that would be a competitive technique to the technique presented here. I would have liked to see those results. The authors have addressed the reviewers comments/questions in detail and I thank them for that.

**Justification For Why Not Higher Score:**

It isn't clear why this method should be used when open-source LLMs could do this task very well. If we are assuming than an NMT exists for these languages then there is a very good chance that an existing LLM can do this as well.

**Justification For Why Not Lower Score:**

The experiments are thorough and sound.

---

### Decision · Program_Chairs · 2024-01-16

Accept (poster)